# ONE-SHOT ACTIVE LEARNING BASED ON LEWIS WEIGHT SAMPLING FOR MULTIPLE DEEP MODELS

**Sheng-Jun Huang**[1]**, Yi Li**[2]**, Yiming Sun**[2] **& Ying-Peng Tang**[1]*

College of Computer Science and Technology, Nanjing University of Aeronautics and Astronautics[1]
School of Physical and Mathematical Sciences, Nanyang Technological University[2]
{huangsj, tangyp}@nuaa.edu.cn  {yili, yiming005}@ntu.edu.sg

## ABSTRACT

Active learning (AL) for multiple target models aims to reduce labeled data querying while effectively training multiple models concurrently. Existing AL algorithms often rely on iterative model training, which can be computationally expensive, particularly for deep models. In this paper, we propose a one-shot AL method to address this challenge, which performs all label queries without repeated model training. Specifically, we extract different representations of the same dataset using distinct network backbones, and actively learn the linear prediction layer on each representation via an $\ell_p$-regression formulation. The regression problems are solved approximately by sampling and reweighting the unlabeled instances based on their maximum Lewis weights across the representations. An upper bound on the number of samples needed is provided with a rigorous analysis for $p \in [1, +\infty)$. Experimental results on 11 benchmarks show that our one-shot approach achieves competitive performances with the state-of-the-art AL methods for multiple target models.

## 1 INTRODUCTION

The rapid advancements in deep learning have led to a substantial increase in demand for extensive labeled data points to effectively train high-performance models. However, data labeling remains costly due to its reliance on human labor. To address this challenge, active learning (AL) (Settles, 2009; Ren et al., 2021) has emerged as an effective strategy for mitigating annotation expenses. This approach estimates the potential utility of different unlabeled instances in improving the performance of a target model and selectively queries the labels of the most beneficial instances from the oracle (i.e., an expert who can provide the ground-truth label). A typical practice of AL conducts label querying and model updating iteratively to exploit the insights from model decisions, i.e., selecting one or a small batch of instances based on the model predictions and updating the target model in each iteration until the labeling budget is exhausted (Cohn et al., 1994; Huang et al., 2014). This paradigm has been widely applied in real-world scenarios (Hoi et al., 2008; Shi & Zhou, 2023).

Recently, there has been a significant surge in the demand for the deployment of machine learning systems on diverse resource-constrained devices (Deng et al., 2020; Gou et al., 2021; Menghani, 2023). For example, speech recognition and face recognition systems usually need to support various types of machines with varying computing and memory resources. As a result, the task of training multiple models with varying complexities using the same labeled dataset has arisen (Cai et al., 2019), leading to a new setting of AL where there are multiple target models to be learned simultaneously (Tang & Huang, 2022).

Tang & Huang (2022) provide both theoretical and empirical evidence showcasing the potential of AL in alleviating the substantial data labeling burden associated with training multiple target models. They propose an iterative AL algorithm DIAM and validate its effectiveness for multiple deep models. However, the use of iterative AL methods results in a significant increase in model training cost. This is due to the requirement of training multiple deep models at each query iteration. A potential solution is increasing the querying batch size of conventional batch-mode AL methods.

---

*All authors contributed equally to this work. Ying-Peng Tang is the corresponding author.

Nevertheless, this may lead to redundant querying (Yang & Loog, 2019). A more cost-effective strategy could be one-shot or single-shot querying, which selects the required number of unlabeled instances and makes all label queries within one iteration devoid of re-training the models.

Most existing one-shot AL methods query a representative set of instances using the distance between feature vectors (Yang & Loog, 2019; Viering et al., 2019; Jin et al., 2022; Shoham & Avron, 2023). However, this approach faces challenges when handling multiple deep models, as the same instance can exhibit different feature representations in different models. This phenomenon arises due to the intrinsic representation learning of deep models, where data representations are implicitly optimized during the training process and varied network architectures yield distinct embeddings. These embeddings may contain abundant information to facilitate data selection. However, such information has not been well exploited by existing one-shot AL methods. Therefore, they may not yield optimal performances in the setting of multiple models.

In this paper, we propose a one-shot AL method for multiple deep models, accompanied by a rigorous theoretical analysis. Our method is based on the fact that a deep model can be viewed as a linear prediction layer (i.e., multiple neuron models) and a nonlinear feature extractor (i.e., the network backbone). Therefore, training multiple deep models can be described as learning linear prediction layers from the outputs of distinct network backbones. In this way, active learning from diverse data representations can be formulated as optimizing a shared sampling matrix to minimize the error of each linear predictor. To facilitate computing and analysis, we consider the learning of the prediction layer as an $\ell_p$ regression problem with $p \in (0, +\infty)$. Notably, our empirical studies place particular emphasis on the case of $p = 2$, i.e., the squared loss, which is one of the most commonly used loss functions in deep learning. Specifically, suppose that there are $k$ models and $A^j \in \mathbb{R}^{n \times d}$ ($j = 1, \ldots, k$) is the feature matrix obtained by feeding the dataset into the $j$-th network backbone. Let $f : \mathbb{R} \to \mathbb{R}$ be an $L$-Lipschitz function with $f(0) = 0$. Typical choices of $f$ are activation functions such as ReLU, Sigmoid, and so on. We abuse the notation and apply $f$ to a vector $\boldsymbol{v} \in \mathbb{R}^n$ coordinatewise, i.e. $f(\boldsymbol{v}) = (f(\boldsymbol{v}_1), \ldots, f(\boldsymbol{v}_n))^T$. Suppose that $\boldsymbol{y}^1, \ldots, \boldsymbol{y}^c \in \mathbb{R}^n$ are $c$ label vectors and the task is to minimize the loss $\sum_{i=1}^c \|f(A^j \boldsymbol{\theta}^{ij}) - \boldsymbol{y}^i\|_p^p$ over $\boldsymbol{\theta}^{1j}, \ldots, \boldsymbol{\theta}^{cj} \in \mathbb{R}^d$ for all models $j$ simultaneously. Since the construction of $S$ is independent of $\boldsymbol{y}^1, \ldots, \boldsymbol{y}^c$, we henceforth assume that $c = 1$, with a single label vector $\boldsymbol{y} \in \mathbb{R}^n$. Therefore, we seek a shared reweighted sampling matrix $S$ such that we can, from the labels of the sampled instances $S\boldsymbol{y}$, approximately solve the regression problem $\min_{\boldsymbol{\theta}} \|f(A^j \boldsymbol{\theta}) - \boldsymbol{y}\|_p^p$ for all models $j$ simultaneously.

The simplest case is when there is a single model, i.e., $k = 1$. In this case, Gajjar et al. (2023a) are the first to study the problem of actively learning a single neuron model. They cast the problem as a least-squares regression problem (i.e. $p = 2$) $\min_{\boldsymbol{\theta}} \|f(A\boldsymbol{\theta}) - \boldsymbol{y}\|_2^2$ and find an $\tilde{\boldsymbol{\theta}}$ such that

$$\|f(A\tilde{\boldsymbol{\theta}}^j) - \boldsymbol{y}\|_2^2 \leq C \cdot \left(\|f(A\boldsymbol{\theta}^*) - \boldsymbol{y}\|_2^2 + \epsilon L^2 \|A\boldsymbol{\theta}^*\|_2^2\right),$$

where $\boldsymbol{\theta}^* = \arg\min_{\boldsymbol{\theta}} \|f(A\boldsymbol{\theta}) - \boldsymbol{y}\|_2^2$ is the minimizer, $C$ is an absolute constant and $\epsilon$ is an accuracy parameter. Recall that $L$ is the Lipschitz constant of $f$. Gajjar et al. (2023a) also show that the additive term $\epsilon L^2 \|A\boldsymbol{\theta}^*\|_2^2$ is necessary. For $k > 1$ and general $p$, we seek approximate solutions $\tilde{\boldsymbol{\theta}}_1, \ldots, \tilde{\boldsymbol{\theta}}_k$ with the following error guarantee of a similar form on each individual model:

$$\|f(A^j \tilde{\boldsymbol{\theta}}^j) - \boldsymbol{y}\|_p^p \leq C \cdot \left(\|f(A^j \boldsymbol{\theta}^j) - \boldsymbol{y}\|_p^p + \epsilon L^p \|A^j \boldsymbol{\theta}^j\|_p^p\right), \tag{1}$$

where $\boldsymbol{\theta}^j = \arg\min_{\boldsymbol{\theta}} \|f(A^j \boldsymbol{\theta}) - \boldsymbol{y}\|_p^p$ is the minimizer for model $j$ and $C = C(p) > 0$ is a constant depending only on $p$.

Gajjar et al. (2023a) construct $S$ to be a leverage score sampling matrix and solve $\tilde{\boldsymbol{\theta}} = \arg\min_{\boldsymbol{\theta} \in E} \|f(SA\boldsymbol{\theta}) - S\boldsymbol{y}\|_2^2$ with $E = \{\boldsymbol{\theta} : \|SA\boldsymbol{\theta}\|_2^2 \leq \|S\boldsymbol{y}\|_2^2/(\epsilon L^2)\}$. At the core of their argument lies the classical fact that such an $S$ gives an $\ell_2$ subspace embedding for $A$, i.e., $\|SA\boldsymbol{\theta}\|_2 \approx \|A\boldsymbol{\theta}\|_2$ for all $\boldsymbol{\theta}$ simultaneously. In fact, it is not necessary to sample the rows of $A$ according to the exact leverage scores $\tau_1(A), \ldots, \tau_n(A)$; any sampling probability proportional to $t_i \gtrsim \tau_i(A)$ for $i$-th row will suffice, with the number of samples being proportional to $\sum_i t_i$. This very fact motivates us to tackle the task of data selection from diverse representations by sampling the rows according to the maximum of leverage scores across $A^j$'s, i.e., letting $t_i \sim \max_j \tau_i(A^j)$. Solving for each model $j$ by $\tilde{\boldsymbol{\theta}}^j = \arg\min_{\boldsymbol{\theta} \in E^j} \|f(SA^j\boldsymbol{\theta}) - S\boldsymbol{y}\|_2^2$ with $E^j = \{\boldsymbol{\theta} : \|SA^j\boldsymbol{\theta}\|_2^2 \leq \|S\boldsymbol{y}\|_2^2/(\epsilon L^2)\}$ will then achieve (1) for $p = 2$. This indicates that the queried instances are effective in learning each of the linear predictors, which fits our problem well. A potential caveat is that the number of samples needed will be proportional to

$\sum_i t_i \sim \sum_i \max_j \tau_i(A^j)$, which could be as large as $kd$. However, empirical studies show that this is not the case for real-world datasets (see Section 3.2) and our approach will thus be efficient.

For general $p$, instead of leverage scores, it is natural to consider Lewis weights, which can be seen as generalizations of leverage scores for general $p$ (see Section 3.1 for the definition). It is known that an $\ell_p$ Lewis weight sampling matrix $S$ give an $\ell_p$ subspace embedding, i.e., $\|SA\boldsymbol{\theta}\|_p \approx \|A\boldsymbol{\theta}\|_p$ for all $\boldsymbol{\theta}$ simultaneously (Cohen & Peng, 2015). The approach mentioned above extends to general $p$ naturally, attaining (1) for general $p$, by sampling according to the maximum Lewis weights and solving an $\ell_p$-regression problem for $\tilde{x}^j$ with an $\ell_p$-version of $E^j$.

**Theoretical Results.** For $k = 1$, the latest result is to use $\tilde{O}(d/\epsilon^4)$ queries Gajjar et al. (2023b), with an analysis specific to $p = 2$. We generalize the approach to the $\ell_p$ Lewis weight sampling for $p \geq 1$ and extends it to $k \geq 1$, giving the following theorem.

**Theorem 1.1** (Informal version of Corollary 3.6). *Let $w_1(A^j), \ldots, w_n(A^j)$ denote the Lewis weights of $A^j$ and $T = \sum_{i=1}^{n} \max_{j \in [k]} w_i(A^j)$. Suppose that $T = \mathrm{poly}(d)$. There exists a randomized algorithm which samples*

$$m \lesssim \begin{cases} \epsilon^{-4} T \log d, & p = 1 \\ \epsilon^{-4} T d^{\max\{\frac{p}{2}-1, 0\}} \log^2 d \log(d/\epsilon) & p > 0 \text{ and } p \neq 1 \end{cases}$$

*unlabeled instances and outputs solutions $\tilde{\boldsymbol{\theta}}^1, \ldots, \tilde{\boldsymbol{\theta}}^k \in \mathbb{R}^d$ such that (1) holds for all $j \in [k]$ with probability at least 0.9.*

Note that for a single matrix $A \in \mathbb{R}^{n \times d}$, the sum $T = \sum_i w_i(A) = d$ and so Theorem 1.1 implies a sample complexity of $\tilde{O}(d^{\max\{p/2, 1\}}/\epsilon^4)$, recovering the result in Gajjar et al. (2023b) for $p = 2$.

**Empirical Findings.** Extensive experiments are conducted on 11 classification and regression benchmarks with 50 distinct deep models. In Section 3.2, we empirically observe that the sum of the maximum leverage scores grows very slowly as the number of models increases. This result reveals the strong correlation among the leverage scores of different deep representations, providing a direction for interpreting deep representation learning (Kornblith et al., 2019; Nguyen et al., 2020). In Section 4, we validate the effectiveness of our method with fine-tuning and vanilla learning scenarios of deep models for both the $\ell_2$-regression loss and cross-entropy loss. The results show that our method outperforms other one-shot baselines. Even when comparing with the state-of-the-art iterative AL methods for multiple models, our approach achieves competitive performance.

## 2 RELATED WORK

Active learning has been extensively studied in the past decades (Settles, 2009; Ren et al., 2021). With a limited query budget, many methods try to query the labels of the most useful instances for a target model by designing effective selection criteria, which commonly depend on two notions, informativeness and representativeness. Informativeness-based criteria prefer instances where the target model has a highly uncertain prediction (Lewis & Gale, 1994; Yan & Huang, 2018; Kirsch et al., 2019), while representativeness-based criteria prefer instances which can help reduce the distribution gap between the queried instances and the entire dataset (Dasgupta & Hsu, 2008; Chattopadhyay et al., 2012; Sener & Savarese, 2018). While most existing methods focus on improving the performance of a specific target model, Tang & Huang (2022) extend the setting of AL to multiple target models. In this scenario, the active learner seeks to enhance the performance of every target model simultaneously by selective querying. Their work demonstrates that the query complexity of AL for multiple models can be upper bounded by that of an appropriately designed single model. Based on this insight, they propose an iterative algorithm called DIAM, which queries the labels of the instances located in the joint disagreement regions among multiple models. Although the method is effective, a significant concern is the substantial cost incurred by training multiple deep models at each iteration.

To reduce the computational cost of repetitive model training, one-shot AL algorithms have been proposed to query all useful instances in a single batch, thereby avoiding the need for model updates. Yang & Loog (2019) employ existing AL methods with pseudo-labeling to obtain a candidate set of diverse instances, and select the queries based on the feature distance between unlabeled instances and candidate instances. Viering et al. (2019) select representative data points by the kernelized discrepancy methods, e.g., Maximum Mean Discrepancy (MMD) (Borgwardt et al., 2006), and give

error bounds under different assumptions on data distribution. Jin et al. (2022) propose a one-shot AL method for deep image segmentation. Their approach uses self-supervised learning to obtain more informative representations and selects diverse instances based on clustering results and feature distances. In addition, Coreset (Sener & Savarese, 2018) and Transductive Experimental Design (Yu et al., 2006) are implicit one-shot AL methods. However, all the aforementioned one-shot AL methods cannot handle the distinct representations of multiple deep models.

Although most existing AL methods rely on heuristics lacking theoretical analysis, AL with Lewis weight sampling has been well studied for active $\ell_p$-regression problems $\min_{\boldsymbol{\theta}} \|A\boldsymbol{\theta} - \boldsymbol{y}\|_p$, where the matrix $A \in \mathbb{R}^{n \times d}$ is fully accessible while the label vector $\boldsymbol{y} \in \mathbb{R}^n$ needs to be queried (Chen & Price, 2019; Chen & Derezinski, 2021; Parulekar et al., 2021; Chen et al., 2022; Musco et al., 2022). Provable guarantees are obtained for $(1+\epsilon)$-approximate solutions, i.e., $\|A\boldsymbol{\theta}' - \boldsymbol{y}\|_p \leq (1+\epsilon)\|A\boldsymbol{\theta}^* - \boldsymbol{y}\|_p$, where $\boldsymbol{\theta}'$ is the output of the algorithm and $\boldsymbol{\theta}^*$ the true minimizer. For $p = 1$, Parulekar et al. (2021) show that $O(\epsilon^{-2}d\log(d/(\epsilon\delta)))$ samples suffice. For $p = 2$, Chen & Price (2019) solve the problem optimally with $O(d/\epsilon)$ queries. For $p \in (1, 2)$, Chen & Derezinski (2021) propose the first algorithm to solve the problem with sublinear query complexity, i.e., $O(\epsilon^{-2}d^2 \log d)$. For $p > 2$, Musco et al. (2022) show that $O(\epsilon^{-p}d^{p/2} \log^2 d \log^{p-1}(d/\epsilon))$ queries suffice. Recently, Gajjar et al. (2023a) extend such sampling method to the single neuron model for $p = 2$, which inspires our work. They establish a multiplicative constant-factor error bound of the form (1) using $O(d^2/\epsilon^4)$ samples. This has been further improved to $O(d/\epsilon^4)$ in Gajjar et al. (2023b).

## 3 OUR APPROACH

### 3.1 PRELIMINARIES

**Notation.** Suppose that the dataset has $n$ instances $\boldsymbol{\alpha}_1, \ldots, \boldsymbol{\alpha}_n$ and each $\boldsymbol{\alpha}_i$ has a ground-truth label $y_i$. The given data consist of a small labeled set $\mathcal{L} = \{(\boldsymbol{\alpha}_i, y_i)\}_{i=1}^{n_l}$, used for model initialization, and a large unlabeled set $\mathcal{U} = \{\boldsymbol{\alpha}_{n_l+i}\}_{i=1}^{n_u}$, used for active querying. Here, $n = n_l + n_u$ and it is assumed that $n_l \ll n_u$. A neural network can be viewed as the composition of a network backbone and a linear prediction layer $\boldsymbol{\theta} \in \mathbb{R}^d$ composed by an activation function $f(\cdot)$. The prediction of the network is given by $f(A\boldsymbol{\theta})$, where $A \in \mathbb{R}^{n \times d}$ is the feature matrix obtained by feeding the dataset into the network backbone. Denote by $\boldsymbol{y} \in \mathbb{R}^n$ the corresponding label vector that needs to be queried. In our theoretical analysis, we assume that $d \ll n$, $A$ has full column rank, the network backbone is fixed during the learning of $\boldsymbol{\theta}$ and $f$ is $L$-Lipschtiz continuous with $f(0) = 0$.

We always assume that $p > 0$. The $\ell_p$ norm of a vector $\boldsymbol{\theta}$ is defined to be $\|\boldsymbol{\theta}\|_p = (\sum_{i=1}^n |\boldsymbol{\theta}_i|^p)^{\frac{1}{p}}$, where $\boldsymbol{\theta}_i$ is the $i$-th coordinate of $\boldsymbol{\theta}$. When $p < 1$, this is not a norm, nevertheless, it remains a well-defined quantity and we shall abuse the notation and denote it by $\|\boldsymbol{\theta}\|_p$.

For a matrix $A$, the operator norm of $A$ is defined as $\|A\|_2 = \sup_{\boldsymbol{\theta} \in \mathbb{R}^d \setminus \{0\}} \|A\boldsymbol{\theta}\|_2 / \|\boldsymbol{\theta}\|_2$. For integer $n \geq 1$, we use $[n]$ to denote the set $\{1, 2, \ldots, n\}$. We write $a = (1 \pm \epsilon)b$ if $(1 - \epsilon)b \leq a \leq (1 + \epsilon)b$ and $a \lesssim_{t_1, t_2, \ldots} b$ if there exists a constant $C$ depending only on $t_1, t_2, \ldots$ such that $a \leq Cb$. We also write $a \sim_{t_1, t_2, \ldots} b$ if $a \lesssim_{t_1, t_2, \ldots} b$ and $b \lesssim_{t_1, t_2, \ldots} a$.

**Lewis Weights Sampling.** We shall define the Lewis weights and state a classical result that Lewis weight sampling gives subspace embeddings, which is the starting point of our algorithm.

**Definition 3.1** ($\ell_p$ Lewis Weights). Suppose that $A \in \mathbb{R}^{n \times d}$ and its $i$-th row is $\boldsymbol{a}_i \in \mathbb{R}^d$. The Lewis weights of $A$ are $w_1, \ldots, w_n$ such that $w_i = (\boldsymbol{a}_i^\top (A^\top W^{1 - \frac{2}{p}} A)^{-1} \boldsymbol{a}_i)^{\frac{p}{2}}$, where $W$ is a diagonal matrix with diagonal elements $w_1, w_2, \ldots, w_n$.

We remark that Lewis weights satisfy that $w_i(A) \in [0, 1]$ and $\sum_{i=1}^n w_i(A) = d$. When $p = 2$, Lewis weights are exactly the leverage scores. Next, we define $\ell_p$ subspace embedding and sampling matrix. Then, we state the result that Lewis weight sampling gives subspace embeddings.

**Definition 3.2** ($\ell_p$ Subspace Embedding). Let $\epsilon \in (0, 1)$ be the distortion parameter. A matrix $S \in \mathbb{R}^{m \times n}$ is said to be an $\ell_p$ $\epsilon$-subspace-embedding matrix for $A \in \mathbb{R}^{n \times d}$ if it holds simultaneously for all vectors $\boldsymbol{\theta} \in \mathbb{R}^d$ that $(1 - \epsilon)\|A\boldsymbol{\theta}\|_p \leq \|SA\boldsymbol{\theta}\|_p \leq (1 + \epsilon)\|A\boldsymbol{\theta}\|_p$.

**Definition 3.3** (Sampling Matrix). Suppose that $p_1, \ldots, p_n \geq 0$ such that $p_1 + p_2 + \cdots + p_n = 1$ and $\boldsymbol{e}_1, \ldots, \boldsymbol{e}_n$ are the standard basis vectors of $\mathbb{R}^n$. A matrix $S \in \mathbb{R}^{m \times n}$ is called a reweighted

sampling matrix if the rows of $S$ are i.i.d. copies of random vector $X$, where $X = (mp_j)^{-1/p}e_j^T$ with probability $p_j$, $j = 1, \ldots, n$. The number $m$ of rows in $S$ is called the sample size.

**Lemma 3.4** (Constant-factor Subspace Embedding, (Cohen & Peng, 2015, Theorem 7.1)). *Given $A \in \mathbb{R}^{n \times d}$. Suppose that $t_i \geq \beta w_i$ for all $i \in [n]$, where*

$$\beta \gtrsim_p \begin{cases} \log^3 d + \log \frac{1}{\delta}, & 0 < p < 2, p \neq 1 \\ \log \frac{d}{\delta}, & p = 1, 2 \\ d^{\frac{p}{2}-1}(\log d + \log \frac{1}{\delta}) & 2 < p < \infty \end{cases}$$

*is a sampling parameter. Let $m = \sum_{i=1}^{n} t_i$. If $S \in \mathbb{R}^{m \times n}$ is a reweighted sampling matrix with sampling probability $p_i = \frac{t_i}{m}$ for all $i$, then $S$ is an $\ell_p$ $\frac{1}{2}$-subspace-embedding matrix for $A$ with probability at least $1 - \delta$.*

We note that our main theorem only requires constant-factor subspace embedding property of the sampling matrix $S$ and, therefore, we can ignore the dependence on $\epsilon$ in the bounds for $\ell_p$ subspace embeddings. The case of $p \leq 2$ is proved by Cohen & Peng (2015) and the case of $p > 2$ is originally due to Bourgain et al. (1989).

## 3.2 AN EMPIRICAL OBSERVATION

Recall that the sample size of maximum Lewis weight sampling is proportional to $\sum_i \max_j w_i(A^j)$. We would like first to examine this sum across representations as it will determine the potential query savings. In the following empirical studies, we mainly consider the case of $p = 2$ (i.e., squared loss), where the Lewis weight becomes exactly the leverage score.

We conduct experiments on 11 datasets. Due to the space limitation, the empirical settings, dataset specifications and more results are deferred to Appendix D. We report the results of MNIST, CIFAR-100 and CelebA datasets below in Figure 1. We plot the theoretical upper bound and the exact values of the sum of the maximum leverage score of each instance across different representations.

The results show that the exact sum grows very slowly as the number of models increases in both classification and regression tasks. This suggests highly consistent discrimination power of most instances across different representations, as the leverage score measures how hard an instance can be linearly represented by others. Therefore, our algorithm is cost-effective. Leverage scores also provide a possible direction to interpret the behavior of deep representation learning, as prior works have not discovered any simple form of correlation among the diverse representations obtained by different model architectures (Kornblith et al., 2019; Nguyen et al., 2020).

## 3.3 THE ALGORITHM

Based on our empirical observations, we propose to sample and reweight unlabeled instances based on their maximum Lewis weights across multiple representations. Specifically, given the feature matrices of the labeled and unlabeled instances (denoted by $\{L^j\}_{j=1}^{k}$ and $\{U^j\}_{j=1}^{k}$, respectively), our algorithm begins with calculating the Lewis weights of the unlabeled instances based on each of their feature representations. Next, a normalized maximum Lewis weight among multiple representations for each unlabeled instance is obtained:

$$p_i = \frac{\max_{j \in [k]} w_i(U^j)}{\sum_{i=1}^{n_u} \max_{j \in [k]} w_i(U^j)}, \quad i = 1, \ldots, n_u.$$

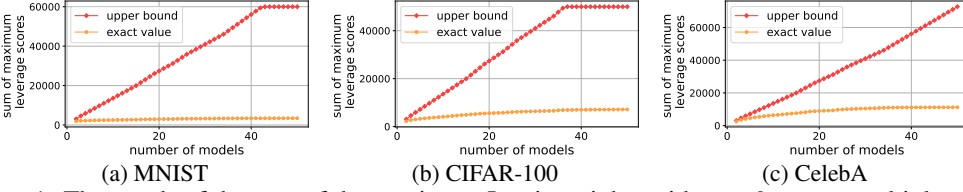

(a) MNIST       (b) CIFAR-100       (c) CelebA

Figure 1: The trends of the sum of the maximum Lewis weights with $p = 2$ among multiple representations as the number of deep models increases.

---

**Algorithm 1** The Proposed Algorithm.

---

**Input:** Feature matrices of labeled and unlabeled instances $L^j, U^j$ ($j = 1, \ldots, k$), query budget $\tau$.
**Output:** Trained linear models $\tilde{\boldsymbol{\theta}}^1, \ldots, \tilde{\boldsymbol{\theta}}^k$.
**Initialize:** $\boldsymbol{p}, \bar{\boldsymbol{y}} \leftarrow$ zero vector of length $n_u$; $\mathcal{Q} \leftarrow$ an empty list; $m \leftarrow 0$
 1: $p_i \leftarrow \max_{1 \leq j \leq k} w_i(U^j)$ for $i = 1, \ldots, n_u$
 2: $p_i \leftarrow p_i/\|\boldsymbol{p}\|_1$ for $i = 1, \ldots, n_u$
 3: **while** $\mathcal{Q}$ has fewer than $\tau$ distinct elements **do**
 4:     $q \leftarrow$ sample a number from $[n_u]$ with replacement with probability $p_1, \ldots, p_{n_u}$
 5:     $m \leftarrow m + 1$
 6:     append $q$ to $\mathcal{Q}$
 7:     **if** the label of $q$-th unlabeled instance is unknown **then**
 8:         $\bar{y}_q \leftarrow$ query the label of $q$-th unlabeled instance
 9: $S \leftarrow$ zero matrix with shape $(n_l + m) \times (n_l + n_u)$
10: $S_{i,i} \leftarrow 1$ for $i = 1, \ldots, n_l$
11: $S_{i+n_l, \mathcal{Q}_i + n_l} \leftarrow (m \cdot p_{\mathcal{Q}_i})^{-1/p}$ for $i = 1, \ldots, m$
12: $\boldsymbol{y} \leftarrow [y_1, \ldots, y_{n_l}, \bar{\boldsymbol{y}}]^T$
13: **for** $j = 1, \ldots, k$ **do**
14:     $A^j \leftarrow \begin{bmatrix} L^j \\ U^j \end{bmatrix}$
15:     $\tilde{\boldsymbol{\theta}}^j \leftarrow \arg\min_{x \in E} \|Sf(A^j\boldsymbol{\theta}) - S\boldsymbol{y}\|_p^p$, where $E = \{\boldsymbol{\theta} : \|SA^j\boldsymbol{\theta}\|_p^p \leq \frac{1}{\epsilon L^p}\|S\boldsymbol{y}\|_p^p\}$
16: **return** $\tilde{\boldsymbol{\theta}}^1, \ldots, \tilde{\boldsymbol{\theta}}^k$

---

In the querying phase, we conduct i.i.d. sampling with replacement on the unlabeled set using a probability distribution $\boldsymbol{p}$. The sampling process is repeated until $\tau$ distinct unlabeled instances are sampled. Let $\mathcal{Q}$ denote the set of indices of unlabeled instances that are selected for label query. We reweight each of the instance with index $q \in \mathcal{Q}$ by $(m \cdot p_q)^{-1/p}$. Finally, both the initially labeled instances with weight $\mathbf{1}$ and the reweighted queried instances will be used to update each of the target model. Note that, although $\mathcal{Q}$ may contain repeated entries, each instance will be queried only once and reoccurrences will not incur additional query cost. We present our algorithm in Algorithm 1.

### 3.4 THEORETICAL GUARANTEES

Our main result is as follows, which can be seen as the guarantee for a single model.

**Theorem 3.5.** *Let $p > 0$, $f(\boldsymbol{\theta})$ be an $L$-Lipschitz function with $f(0) = 0$, $A \in \mathbb{R}^{n \times d}$ be the data matrix and $\boldsymbol{y} \in \mathbb{R}^n$ be the target vector. Consider a reweighted sampling matrix $S$ with with row sampling probability $p_i = \frac{t_i}{m}$, where $t_1, \ldots, t_n$ are some quantities and $m = \sum_i t_i$.*

*Suppose that $t_1, \ldots, t_n \in \mathbb{R}$ satisfy that $t_i \geq \beta w_i(A)$, where*

$$\beta \gtrsim_p \begin{cases} \epsilon^{-4} \log(\sum_{i=1}^n t_i), & p = 1 \\ \epsilon^{-4} d^{\max\{\frac{p}{2}-1, 0\}} \log^2 d \log(\sum_{i=1}^n t_i), & p > 0 \text{ and } p \neq 1. \end{cases} \tag{2}$$

*Then, if $S$ is a reweighted sampling matrix as described above and $\tilde{\boldsymbol{\theta}} = \arg\min_{\boldsymbol{\theta} \in E} \|Sf(Ax) - Sy\|_p$, where $E = \{\boldsymbol{\theta} : \|SA\boldsymbol{\theta}\|_p^p \leq \|S\boldsymbol{y}\|_p^p/(\epsilon L^p)\}$, it holds with probability at least $0.9$ that*

$$\|f(A\tilde{\boldsymbol{\theta}}) - \boldsymbol{y}\|_p^p \leq C \left( \|f(A\boldsymbol{\theta}^*) - \boldsymbol{y}\|_p^p + \epsilon L^p \|A\boldsymbol{\theta}^*\|_p^p \right),$$

*where $\boldsymbol{\theta}^* = \arg\min_{\boldsymbol{\theta}} \|f(A\boldsymbol{\theta}) - \boldsymbol{y}\|_p$ and $C > 0$ is a constant depending only on $p$.*

The proof of Theorem 3.5 is deferred to Appendix A. Our analysis also suggests that an $\ell_p$-subspace-embedding can be obtained using $\tilde{O}(d/\epsilon^2)$ samples, removing the $\log n$ factor in (Woodruff & Yasuda, 2023), which may be of independent interest. See Appendix B for discussions. Below we show the guarantee for multiple models, which follows easily as a corollary of Theorem 3.5.

**Corollary 3.6.** *Let $A_1, \ldots, A_k \in \mathbb{R}^{n \times d}$ be data matrices and $T = \sum_{i=1}^n \max_{j \in [k]} w_i(A^j)$. Let $f(\boldsymbol{\theta})$ be an $L$-Lipschitz function with $f(0) = 0$ and $\boldsymbol{y} \in \mathbb{R}^n$ be the target vector. There exists an algorithm that makes*

$$m \sim_p \begin{cases} \epsilon^{-4} T \log(T/\epsilon), & p = 1 \\ \epsilon^{-4} T d^{\max\{\frac{p}{2}-1, 0\}} \log^2 d \log(dT/\epsilon), & p > 0 \text{ and } p \neq 1 \end{cases} \tag{3}$$

*queries and outputs solutions $\tilde{\boldsymbol{\theta}}^1, \ldots, \tilde{\boldsymbol{\theta}}^k \in \mathbb{R}^d$ such that (1) holds for all $j \in [k]$ with probability at least 0.9.*

*Proof.* Let $t_i = \beta \cdot \max_j w_i(A^j)$, then for any fixed $j$, it holds that $t_i \geq \beta w_i(A^j)$. Also, $m = \sum_i t_i = \beta T$. The sampling probability $p_i = t_i/m = \max_j w_i(A^j)/T$, which is exactly our sampling scheme in Algorithm 1. Take

$$\beta \sim \begin{cases} \epsilon^{-4} \log d, & p = 1 \\ \epsilon^{-4} d^{\max\{\frac{p}{2}-1, 0\}} \log^2 d \log(dT/\epsilon), & p > 0 \text{ and } p \neq 1 \end{cases},$$

then $\beta$ satisfies the condition (2) in Theorem 3.5, whence the conclusion follows. $\qquad \square$

**Remark.** The proof of Corollary 3.6 implies the same guarantee for Algorithm 1 if $\tau$ is set to be the quantity for $m$ in (3). Indeed, the proof of Corollary 3.6 shows that the guarantee holds as soon as the variable $m$ in Algorithm 1 reaches the desired amount in (3), which allows double counting of identical sampled rows; setting $\tau$ to be the same value will only result in a larger number $m$ of samples and the guarantee will persist.

## 4 EXPERIMENT

In this section, we conduct experiments to validate the effectiveness of our method[1]. Due to the space limitation, some empirical settings and experimental results are presented in the appendix.

**Empirical Settings.** We incorporate two learning scenarios in our experiments, i.e., fine-tuning and vanilla deep learning. The first one is a common learning scenario for big models. It first pre-trains the model on preliminary tasks. Then, the weights of the network backbone are fixed, and only the prediction heads are fine-tuned on downstream tasks. This setting aligns well with our problem formulation. The second scenario is the default learning scheme, i.e., updating all the parameters of the network with the training dataset.

We employ 50 distinct network architectures as the target models. These architectures are published by a recent NAS method OFA (Cai et al., 2019) for accommodating diverse resource-constraint devices, ranging from NVIDIA Tesla V100 GPU to mobile devices. It aligns well with our problem setting. We conduct experiments on 11 datasets, including 8 classification benchmarks: MNIST (LeCun et al., 1998), Fashion-MNIST (Xiao et al., 2017), Kuzushiji-MNIST (Clanuwat et al., 2018), SVHN (Netzer et al., 2011), EMNIST-letters and EMNIST-digits (Cohen et al., 2017), CIFAR-10 and CIFAR-100 (Krizhevsky, 2009); and 3 regression benchmarks: Biwi (Fanelli et al., 2013), FLD (Sun et al., 2013) and CelebA (Liu et al., 2015). The specifications of the datasets and model configurations are deferred to the Appendix E.1. The active learning settings are outlined as follows.

- For the scenario of vanilla deep learning, we conduct performance comparisons on the classification benchmarks. Specifically, 3000 instances are sampled uniformly from the training set to initialize the models. The other compared methods will then select 3000 unlabeled instances from the remaining data points for querying at each iteration, while our method conducts one-shot querying with budgets of 9000 and 15000 instances. The cross-entropy loss is employed in model training. In this scenario, the one-shot methods also query 3000 instances per batch for better comparison. However, these methods select batches independently.
- For the fine-tuning scenario, we use the regression datasets. Initially, 500 instances are sampled uniformly from the training set to fine-tune each network. Then, we fix the backbone parameters and actively query the labels among the remaining instances. Afterwards, 50 linear prediction layers with mean squared error (MSE) loss and ReLU activation function are trained on the updated labeled dataset, utilizing the features extracted by different network backbones. In this scenario, all the compared methods have the same query budgets of 3000 and 6000 instances.

We compare our algorithm with the following methods in the vanilla deep learning scenario.

- (iterative) DIAM (Tang & Huang, 2022): The state-of-the-art iterative AL method for multiple target models, which prefers the instances in the joint disagreement regions of multiple models.

---

[1]All experiments are conducted on a machine with four GeForce RTX 3090 graphic cards and an Intel Xeon Gold 5317 CPU. The source code is included in the supplementary material for experiment reproducibility.

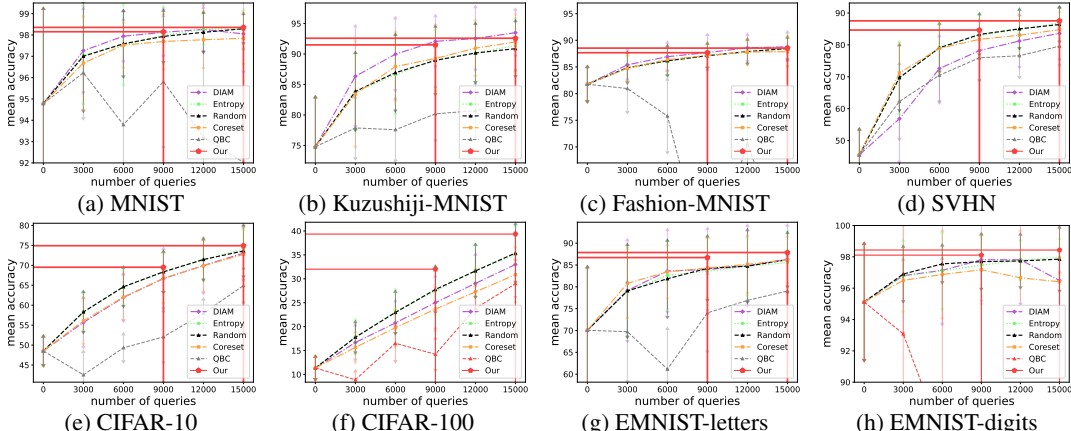

Figure 2: Results of Performance comparison in classification datasets. The error bars indicate the standard deviation of the performances of multiple models.

- (iterative) Entropy (Lewis & Catlett, 1994): This strategy selects instances with the highest prediction entropy. We follow the implementation in (Tang & Huang, 2022) to adapt it to multiple models. It queries the instances with the highest mean prediction entropy.
- (iterative) QBC (Seung et al., 1992): This strategy selects the instances that the target models have the most inconsistent predictions. The inconsistency is evaluated by KL divergence.
- (one-shot) Coreset (Sener & Savarese, 2018): This strategy selects the most representative instances. We follow the implementation in (Tang & Huang, 2022) to adapt it to multiple models. It solves the coreset problem based on the features extracted by the supernet in OFA.
- (one-shot) Random: This strategy selects instances uniformly from the unlabeled pool.

In the fine-tuning scenario, fewer existing methods are available. Specifically, we compare our algorithm with Coreset, Random and QBC methods. Although QBC is usually implemented in an iterative fashion, we employ a large query batch size for it to unify the query settings.

Our method selects and reweights the unlabeled instances based on the leverage scores (i.e., $p = 2$) in both scenarios. Note that, in the fine-tuning scenario, our implementations remove the constraint $E$ in Line 15 in Algorithm 1 for better examination of the practicability. In the vanilla deep learning scenario, we use the default training scheme of deep models to replace Line 15 in Algorithm 1. The mean accuracy and the mean MSE are used to evaluate the performances of multiple target models for classification and regression tasks, respectively.

**Experiment Results.** We report the performance comparison results in Figure 2 and Figure 3. In the scenario of vanilla deep learning, we can observe that our one-shot method achieves comparable performances with the other iterative AL methods in most cases. This phenomenon indicates that our method can significantly reduce the costs of training multiple deep model while preserving its proficiency in the ability of query saving. QBC is the worst one. We find that it causes a severe class imbalance according to the results in Table 5 in the appendix. This may explain its inferior performances. Coreset is usually worse than Random. Note that, the problem settings of Sener & Savarese (2018) and our work are different. there are 50 distinct target networks to be learned in

Table 1: Comparisons on the running time between our method and the other baselines with a query budget 15000 instances. The running time includes data querying and model training (GPU hours).

|         | MNIST  | F.MNIST | K.MNIST | SVHN   | CIF.10 | CIF.100 | EMN.l. | EMN.d.  |
|---------|--------|---------|---------|--------|--------|---------|--------|---------|
| DIAM    | 46.643 | 47.597  | 46.765  | 52.228 | 45.493 | 53.532  | 73.522 | 120.840 |
| QBC     | 23.937 | 24.419  | 24.502  | 26.011 | 25.541 | 30.498  | 36.280 | 40.231  |
| Entropy | 24.060 | 24.293  | 24.455  | 25.792 | 25.173 | 28.655  | 34.291 | 42.719  |
| Our     | 5.299  | 5.366   | 5.354   | 5.605  | 5.350  | 5.57    | 9.717  | 12.711  |
| Coreset | 5.200  | 5.201   | 5.285   | 5.466  | 5.450  | 5.745   | 8.984  | 11.043  |
| Random  | 4.317  | 4.333   | 4.402   | 4.583  | 4.567  | 4.712   | 7.317  | 8.027   |

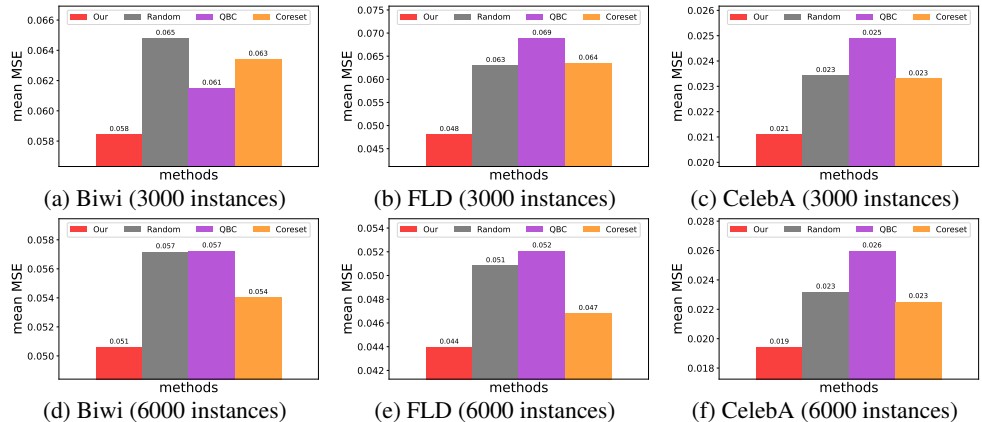

Figure 3: Results of performance comparisons in regression datasets with different query budgets.

our experiment. The Coreset implementation of Tang & Huang (2022) solves the coreset problem based on the features extracted by the supernet. A drawback of this approach is that the selected instances may not be useful for other models, because the data representations are different. We believe this is the reason that why Coreset is less effective than Random in our setting. Entropy method achieves comparable performances with Random. The reason may also be evidenced by the results in Table 5 in the appendix that their class imbalance ratios are highly consistent, implies that the mean entropy scores tend to have an extremely small standard deviation. The performances of DIAM are less stable. It is effective in the datasets associated with MNIST, but fails on the others. This deficiency has not been observed in our method.

In the scenario of fine-tuning, Figure 3 shows that our approach outperforms than the other baselines with different querying budgets in terms of achieving better mean MSE. These results indicate that our method is effective and robust to different query budgets, it can effectively identify the desired number of useful unlabeled instances under diverse representations to learn linear prediction layers.

We further examine the running time of different AL methods. The results are reported in Table 1. For the one-shot methods Coreset and Random, we report their running time of one-shot querying 15000 instances. It can be observed that the cost of repeated model training is prohibitive in AL for multiple deep models, demonstrating the advantages of one-shot querying. Among the active selection methods, DIAM is the slowest approach because it selects instances based on the predictions of the unlabeled dataset in the latter half of training epochs of each target model. Generating the predictions from multiple models could be expensive, particularly with a large unlabeled pool. QBC and Entropy exhibit similar time costs. Both of them need to feed the unlabeled instances into 50 models to obtain their predictions.

In the fine-tuning scenario, all the compared methods conduct one-shot querying and linear prediction layers are trained with the same computational costs. As a result, the running time of the compared methods is comparable. The results are deferred to Table 6 in the appendix.

## 5 CONCLUSION

In this paper, we propose a one-shot AL algorithm for multiple deep models. The task is formulated as seeking a shared reweighted sampling matrix to approximately solve multiple $\ell_p$-regression problems for neuron models on distinct deep representations. Our approach is to sample and reweight the unlabeled instances based on their maximum Lewis weights across different representations. We establish an upper bound on the number of samples needed by our algorithm to achieve constant-factor approximations for multiple models and general $p$. Our techniques on the one hand substantially improve the upper bound on the number of samples of (Gajjar et al., 2023a) in the case of single model and $p = 2$, on the other hand remove the $\log n$ factor in (Woodruff & Yasuda, 2023) for Lewis weight sampling to obtain $\ell_p$-subspace-embedding. Extensive experiments are conducted on 11 benchmarks and 50 deep models. We observe that the sum of the maximum Lewis weights with $p = 2$ grows very slowly as the number of target models increases, providing a direction for interpreting deep representation learning. The performance comparisons show that our algorithm achieves competitive performances with the state-of-the-art AL methods for multiple deep models.

## ACKNOWLEDGMENTS

S.-J. Huang is supported in part by the National Science and Technology Major Project (2020AAA0107000), the Natural Science Foundation of Jiangsu Province of China (BK20222012, BK20211517), and NSFC (62222605). Y. Li is supported in part by the Singapore Ministry of Education (AcRF) Tier 2 grant MOE-T2EP20122-0001 and Tier 1 grant RG75/21. Y.-P. Tang was supported in part by the China Scholarship Council during his visit to Nanyang Technological University, where most of this work was done.

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

## A    PROOF OF THEOREM 3.5

We first need a simple inequality.

**Fact A.1.** Suppose that $a, b > 0$ and $p > 0$. It holds that $(a + b)^p \leq 2^{|p-1|}(a^p + b^p)$.

Let $\text{OPT} = \min_{\boldsymbol{\theta}} \|A\boldsymbol{\theta} - \boldsymbol{y}\|_p$. Theorem 3.5 is proved by the following chain of inequalities.

$$\left\| f(A\tilde{\boldsymbol{\theta}}) - \boldsymbol{y} \right\|_p^p \overset{\text{(A)}}{\leq} 2^{|p-1|}(\left\| f(A\tilde{\boldsymbol{\theta}}) - f(A\boldsymbol{\theta}^*) \right\|_p^p + \text{OPT}^p)$$

$$\overset{\text{(B)}}{\leq} 2^{|p-1|}(\left\| Sf(A\tilde{\boldsymbol{\theta}}) - Sf(A\boldsymbol{\theta}^*) \right\|_p^p + \epsilon^2 L^p R^p + \text{OPT}^p)$$

$$\overset{(C)}{\leq} 2^{|p-1|}(2^{|p-1|}\left\|Sf(A\tilde{\boldsymbol{\theta}}) - S\boldsymbol{y}\right\|_p^p + C_1\text{OPT}^p + \epsilon^2 L^p R^p)$$

$$\overset{(D)}{\leq} 2^{|p-1|}\left[C_2\left(\text{OPT}^p + \epsilon L^p\left\|A\boldsymbol{\theta}^*\right\|_p^p\right) + C_1\text{OPT}^p + \epsilon^2 L^p R^p\right]$$

$$\overset{(E)}{\leq} C(\text{OPT}^p + \epsilon L^p\left\|A\boldsymbol{\theta}^*\right\|_p^p)$$

where inequalities (A) and (C) use Fact A.1, inequality (D) uses (Gajjar et al., 2023a, Claim 1). Inequality (E) follows from that

$$R^p := \max(\|A\tilde{\boldsymbol{\theta}}^p\|, \|A\boldsymbol{\theta}^*\|^p) \leq \|A\tilde{\boldsymbol{\theta}}\|^p + \|A\boldsymbol{\theta}^*\|^p$$

$$\overset{(EA)}{\leq} 2\left\|SA\tilde{\boldsymbol{\theta}}\right\|_p^p + \|A\boldsymbol{\theta}^*\|_p^p$$

$$\overset{(EB)}{\leq} 2\frac{\|S\boldsymbol{y}\|_p^p}{\epsilon L^p} + \|A\boldsymbol{\theta}^*\|_p^p$$

$$\overset{(EC)}{\leq} 100\frac{\|\boldsymbol{y}\|_p^p}{\epsilon L^p} + \|A\boldsymbol{\theta}^*\|_p^p$$

$$\overset{(ED)}{\leq} 100 \cdot 2^{|p-1|}\frac{\|f(A\boldsymbol{\theta}^*) - y\|_p^p + L^p\|A\boldsymbol{\theta}^*\|_p^p}{\epsilon L^p} + \|A\boldsymbol{\theta}^*\|_p^p$$

$$= 100 \cdot 2^{|p-1|}\frac{\|f(A\boldsymbol{\theta}^*) - y\|_p^p}{\epsilon L^p} + \left(\frac{100 \cdot 2^{|p-1|}}{\epsilon} + 1\right)\|A\boldsymbol{\theta}^*\|_p^p,$$

where inequality (EA) holds because $S$ is a subspace embedding matrix for $A$, inequality (EB) is from the constraint of our approximate solution in Line 16, inequality (EC) holds with probability at least $49/50$ by Markov's inequality and inequality (ED) follows from Fact A.1.

We shall prove inequality (B) in the following lemma. We note that the following lemma is proved in (Gajjar et al., 2023a, Lemmata 2 and 3), but their sampling complexity is $\tilde{O}(d^2/\epsilon^4)$ with an additional $d$ factor compared with ours. We improve their result by using the reduction technique and removing the $\epsilon$-net argument.

**Lemma A.2.** *Suppose that $A \in \mathbb{R}^{n \times d}$ and $t_1, \dots, t_n \in \mathbb{R}$ such that $t_i \geq \beta w_i(A)$ for all $i$ and $p \geq 1$. Let $m = \sum_i t_i$ and $S \in \mathbb{R}^{m \times n}$ be a reweighted sampling matrix of with row sampling probabilities $p_1, \dots, p_n$, where $p_i = \frac{t_i}{m}$. If*

$$\beta \gtrsim \frac{d^{\max\{\frac{p}{2}-1,0\}}}{\epsilon^2}\left(\log^2 d \log m + \log\frac{1}{\delta}\right)$$

*then with probability at least $1 - \delta$ and fixed constant $R > 0$, it holds for all pairs of vectors $\boldsymbol{\theta}_1, \boldsymbol{\theta}_2 \in \mathbb{R}^d$ with $\|A\boldsymbol{\theta}_1\|_p \leq R$ and $\|A\boldsymbol{\theta}_2\|_p \leq R$ that*

$$\|Sf(A\boldsymbol{\theta}_1) - Sf(A\boldsymbol{\theta}_2)\|_p^p = \|f(A\boldsymbol{\theta}_1) - f(A\boldsymbol{\theta}_2)\|_p^p \pm \epsilon L^p R^p.$$

*Proof.* Let $\boldsymbol{x} = f(A\boldsymbol{\theta}_1) - f(A\boldsymbol{\theta}_2)$ and $\boldsymbol{y} = A\boldsymbol{\theta}_1 - A\boldsymbol{\theta}_2$. Denote $T$ to be the set $\mathcal{B}(R) \times \mathcal{B}(R) = \{(\boldsymbol{\theta}_1, \boldsymbol{\theta}_2) : \|SA\boldsymbol{\theta}_1\|_p \leq R, \|SA\boldsymbol{\theta}_2\|_p \leq R\}$. We shall try to upper bound

$$\underset{S}{\mathbb{E}}\left(\max_{(\boldsymbol{\theta}_1, \boldsymbol{\theta}_2)\in T}\left|\|S\boldsymbol{x}\|_p^p - \|\boldsymbol{x}\|_p^p\right|\right)^{\ell}$$

for $\ell = \log\frac{1}{\delta}$.

Since taking the $\ell$-th moment of the maximum is a convex function and $\mathbb{E}\|S\boldsymbol{x}\|_p^p = \|\boldsymbol{x}\|_p^p$, the symmetrization trick yields that

$$\underset{S}{\mathbb{E}}\left(\max_{(\boldsymbol{\theta}_1, \boldsymbol{\theta}_2)\in T}\left|\|S\boldsymbol{x}\|_p^p - \|\boldsymbol{x}\|_p^p\right|\right)^{\ell} \leq 2^{\ell}\underset{S,\sigma}{\mathbb{E}}\left(\max_{(\boldsymbol{\theta}_1, \boldsymbol{\theta}_2)\in T}\left|\sum_{k=1}^{m}\sigma_k\frac{|x_{i_k}|^p}{mp_{i_k}}\right|\right)^{\ell},$$

where $\sigma_k$'s are Rademacher variables.

It follows from Lemma 3.4 that $S$ is a $\frac{1}{2}$-subspace embedding matrix of $A$ with probability at least $1 - \delta/2$. Furthermore, by Lemma A.4, with probability at least $1 - \delta/2$, the Lewis weights of $SA$ is upper bounded by $\frac{1}{\beta}$. Let $\mathcal{E}$ denote the event on $S$ that the above two conditions hold. Then $\Pr(\mathcal{E}) \geq 1 - \delta$. We assume the following proof is conditioned on $\mathcal{E}$.

Next, we prove the conditional expectation over $S$ and $\sigma$ when conditioned on $\mathcal{E}$ satisfies that

$$\mathop{\mathbb{E}}_{S,\sigma}\left[\left(\max_{(\boldsymbol{\theta}_1,\boldsymbol{\theta}_2)\in T}\left|\sum_{k=1}^{m}\sigma_k\frac{|x_{i_k}|^p}{mp_{i_k}}\right|\right)^{\ell}\middle|\mathcal{E}\right]\leq\left(\frac{\epsilon}{2}L^pR^p\right)^{\ell}\delta. \tag{4}$$

Once (4) is established, it would follow Markov's inequality that

$$\Pr\left\{\max_{(\boldsymbol{\theta}_1,\boldsymbol{\theta}_2)\in T}\left|\|S\boldsymbol{x}\|_p^p-\|\boldsymbol{x}\|_p^p\right|\geq\epsilon L^pR^p\middle|\mathcal{E}\right\}$$

$$\leq\frac{\mathbb{E}_{S,\sigma}[\left(\max_{(\boldsymbol{\theta}_1,\boldsymbol{\theta}_2)\in T}\left|\|S\boldsymbol{x}\|_p^p-\|\boldsymbol{x}\|_p^p\right|\right)^{\ell}|\mathcal{E}]}{(\epsilon L^pR^p)^{\ell}}$$

$$\leq2^{\ell}\frac{\mathbb{E}_{S,\sigma}\left[\left(\max_{(\boldsymbol{\theta}_1,\boldsymbol{\theta}_2)\in T}\left|\sum_{k=1}^{m}\sigma_k\frac{|x_{i_k}|^p}{mp_{i_k}}\right|\right)^{\ell}\middle|\mathcal{E}\right]}{(\epsilon L^pR^p)^{\ell}}$$

$$\leq2^{\ell}\frac{(\frac{\epsilon}{2}L^pR^p)^{\ell}\delta}{(\epsilon L^pR^p)^{\ell}}\qquad\text{(by (4))}$$

$$=\delta.$$

and then a union bound that

$$\Pr\left\{\max_{(\boldsymbol{\theta}_1,\boldsymbol{\theta}_2)\in T}\left|\|S\boldsymbol{x}\|_p^p-\|\boldsymbol{x}\|_p^p\right|\geq\epsilon L^pR^p\middle|\mathcal{E}\right\}<2\delta,$$

which would complete the proof after rescaling $\delta$ to $\delta/2$.

Now we focus on the proof of (4), which mostly follows the same approach of Theorem 15.13 in Ledoux & Talagrand (1991).

Let

$$\boldsymbol{u}_k=\frac{f(\boldsymbol{a}_{i_k}^\top\boldsymbol{\theta}_1)-f(\boldsymbol{a}_{i_k}^\top\boldsymbol{\theta}_2)}{(mp_{i_k})^{1/p}},\quad\boldsymbol{v}_k=\frac{\boldsymbol{a}_{i_k}^\top\boldsymbol{\theta}_1-\boldsymbol{a}_{i_k}^\top\boldsymbol{\theta}_2}{(mp_{i_k})^{1/p}},\quad k\in[m].$$

Then $\boldsymbol{u}=S\boldsymbol{x}$ and $\boldsymbol{x}=S\boldsymbol{y}$. We also denote

$$\Lambda=\max_{(\boldsymbol{\theta}_1,\boldsymbol{\theta}_2)\in T}\left|\sum_{k=1}^{m}\sigma_k|\boldsymbol{u}_k|^p\right|,$$

so (4) can be rewritten as

$$\mathop{\mathbb{E}}_{S,\sigma}\left[\Lambda^{\ell}\middle|\mathcal{E}\right]\leq\left(\frac{\epsilon}{2}L^pR^p\right)^{\ell}\delta.$$

We shall split the sum in $\Lambda$ into two parts: large Lewis weights and small Lewis weights. Specifically, we define $\lambda_k=w_k(SA)/d$ to be the reweighted Lewis weight of $SA$ and $J=\{k\in[m]:\lambda_k\geq\frac{1}{m^2}\}$.

First consider those coordinates not in $J$ (small Lewis weights).

$$\max_{(\boldsymbol{\theta}_1,\boldsymbol{\theta}_2)\in T}\left|\sum_{k=1,k\notin J}^{m}\sigma_k|\boldsymbol{u}_k|^p\right|\leq\sum_{k\notin J}|\boldsymbol{u}_k|^p\leq L^p\sum_{k\notin J}\lambda_k|\lambda_k^{-\frac{1}{p}}\boldsymbol{v}_k|^p\leq\frac{2^p}{m}d^{\max(1,\frac{p}{2})}L^pR^p,$$

where the last inequality follows from the fact (see (Ledoux & Talagrand, 1991, Lemma 15.17)) that

$$\max_{k\in[m]}|\lambda_k^{-\frac{1}{p}}\boldsymbol{v}_k|\leq d^{\max(\frac{1}{p},\frac{1}{2})}\|\boldsymbol{v}\|_p \tag{5}$$

and (by the definition of $\mathcal{B}(R)$) that $\|\boldsymbol{v}\|_p\leq2R$.

Next we consider the coordinates in $J$ (large Lewis weights). We have

$$\max_{(\boldsymbol{\theta}_1,\boldsymbol{\theta}_2)\in T}\left|\sum_{k\in J}\sigma_k|\boldsymbol{u}_k|^p\right|=\max_{(\boldsymbol{\theta}_1,\boldsymbol{\theta}_2)\in T}\left|\sum_{k\in J}\lambda_k\sigma_k|\lambda_k^{-\frac{1}{p}}\boldsymbol{u}_k|^p\right|$$

$$\leq\sqrt{\frac{1}{d\beta}}\max_{(\boldsymbol{\theta}_1,\boldsymbol{\theta}_2)\in T}\left|\sum_{k\in J}\sqrt{\lambda_k}\sigma_k|\lambda_k^{-\frac{1}{p}}\boldsymbol{u}_k|^p\right|,$$

where the second line follows from the fact that reweighted Lewis weights of $SA$ are upper bounded by $\frac{1}{d\beta}$. By the triangle inequality, we have

$$\mathbb{E}_{\sigma}[\Lambda^{\ell}|\mathcal{E}] \leq \left(\frac{2^p}{m}d^{\max(1,\frac{p}{2})}L^p R^p\right)^{\ell} + (\frac{1}{d\beta})^{\frac{\ell}{2}} \mathbb{E}_{\sigma}\left[\max_{(\boldsymbol{\theta}_1,\boldsymbol{\theta}_2)\in T}\left|\sum_{k\in J}\sqrt{\lambda_k}\sigma_k|\lambda_k^{-\frac{1}{p}}\boldsymbol{u}_k|^p\right|^{\ell}\middle|\mathcal{E}\right]$$

$$=: \left(\frac{2^p}{m}d^{\max(1,\frac{p}{2})}L^p R^p\right)^{\ell} + (\frac{1}{d\beta})^{\frac{\ell}{2}}\mathbb{E}_{\sigma}[\Xi^{\ell}|\mathcal{E}],$$

where

$$\Xi = \max_{(\boldsymbol{\theta}_1,\boldsymbol{\theta}_2)\in T}\left|\sum_{k\in J}\sqrt{\lambda_k}\sigma_k|\lambda_k^{-\frac{1}{p}}\boldsymbol{u}_k|^p\right|.$$

To bound $\mathbb{E}_{\sigma}[\Xi^{\ell}|\mathcal{E}]$, we introduce the associated distance $\delta((\boldsymbol{\theta}_1,\boldsymbol{\theta}_2),(\boldsymbol{\theta}_1',\boldsymbol{\theta}_2'))$ so that it is enough to bound it by the estimated entropy of $\mathcal{B}(R)$. We define the distance to be

$$\delta^2((\boldsymbol{\theta}_1,\boldsymbol{\theta}_2),(\boldsymbol{\theta}_1',\boldsymbol{\theta}_2'))$$

$$= \sum_{k\in J}\lambda_k\left(\frac{\left|\lambda_k^{-\frac{1}{p}}[f(\boldsymbol{a}_{i_k}^{\top}\boldsymbol{\theta}_1) - f(\boldsymbol{a}_{i_k}^{\top}\boldsymbol{\theta}_2)]\right|^p}{mp_{i_k}} - \frac{\left|\lambda_k^{-\frac{1}{p}}[f(\boldsymbol{a}_{i_k}^{\top}\boldsymbol{\theta}_1') - f(\boldsymbol{a}_{i_k}^{\top}\boldsymbol{\theta}_2')]\right|^p}{mp_{i_k}}\right)^2 \quad (6)$$

$$:= \sum_{k\in J}\lambda_k(|\lambda_k^{-\frac{1}{p}}\boldsymbol{u}_k|^p - |\lambda_k^{-\frac{1}{p}}\boldsymbol{u}_k'|^p)^2$$

and the norm

$$\|\theta\|_J := \max_{k\in J}\frac{|\lambda_k^{-\frac{1}{p}}\boldsymbol{a}_{i_k}^{\top}\theta|}{(mp_{i_k})^{\frac{1}{p}}}. \quad (7)$$

By the tail bound of Dudley's integral (see e.g. (Vershynin, 2018, Theorem 8.1.6)), it holds that

$$\Pr\left\{\Xi \gtrsim \int_0^{\infty}(\log N(T,\delta,\epsilon))^{\frac{1}{2}}d\epsilon + z\cdot\text{diam}(T)\middle|\mathcal{E}\right\} \leq \exp(-z^2).$$

According to Lemma A.3, it holds that

$$\int_0^{\infty}(\log N(T,\delta,\epsilon))^{\frac{1}{2}}d\epsilon \lesssim d^{\max(\frac{p-2}{4},0)}L^p R^{p-1}\int_0^{\infty}(\log N(\mathcal{B}(R),B_J,\epsilon))^{\frac{1}{2}}d\epsilon.$$

For $p \geq 2$, the entropy estimate in (Ledoux & Talagrand, 1991, Proposition 15.18) gives that

$$d^{\frac{p-2}{4}}L^p R^{p-1}\int_0^{\infty}(\log N(\mathcal{B}(R),B_J,\epsilon))^{\frac{1}{2}}d\epsilon$$

$$= d^{\frac{p-2}{4}}L^p R^{p-1}\int_0^{\infty}(\log N(\mathcal{B}(1),B_J,\frac{\epsilon}{R}))^{\frac{1}{2}}d\epsilon$$

$$\lesssim d^{\frac{p-2}{4}}L^p R^{p-1}\left(\int_0^1\left(d\log\left(1+\frac{R\sqrt{d}}{\epsilon}\right)\right)^{\frac{1}{2}}d\epsilon + \int_1^{2\sqrt{d}}\left(\frac{R^2}{\epsilon^2}d\log m\right)^{\frac{1}{2}}d\epsilon\right)$$

$$\lesssim d^{\frac{p}{4}}L^p R^p \log d\sqrt{\log m}.$$

For $1 < p \leq 2$, it follows from the entropy estimate in (Ledoux & Talagrand, 1991, Proposition 15.19) and a similar argument to that for $p \geq 2$ that

$$\int_0^{\infty}(\log N(T,\delta,\epsilon))^{\frac{1}{2}}d\epsilon \lesssim d^{\frac{1}{2}}L^p R^p \log d\sqrt{\log m}.$$

By the property of subgaussian variables (see e.g. (Chen et al., 2022, Proposition 4.12)), we have

$$\mathbb{E}_{\sigma}[\Xi^{\ell}|\mathcal{E}] \leq K^{\ell}(\sqrt{\ell}d^{\max\{\frac{p}{4},\frac{1}{2}\}}L^p R^p + d^{\max(\frac{p}{4},\frac{1}{2})}L^p R^p \log d\sqrt{\log m})^{\ell}.$$

Hence, given $\ell = \log(1/\delta)$, as long as $\beta \geq 2^{p+1}e\cdot\epsilon^{-2}K^2 d^{\max(\frac{p}{2}-1,0)}(\log(1/\delta) + \log^2 d\log m)$, it follows that

$$\mathbb{E}_{\sigma}[\Lambda^{\ell}|\mathcal{E}] \leq \left(\frac{2^p}{m}d^{\max(1,\frac{p}{2})}L^p R^p\right)^{\ell} + (\frac{1}{d\beta})^{\frac{\ell}{2}}\mathbb{E}_{\sigma}[\Xi^{\ell}|\mathcal{E}]$$

$$\leq \left( \frac{2^p}{d\beta} d^{\max(1,\frac{p}{2})} L^p R^p \right)^\ell + \left( \frac{Kd^{\max(\frac{p}{4},\frac{1}{2})} L^p R^p (\sqrt{\ell} + \log d \sqrt{\log m})}{\sqrt{d\beta}} \right)^\ell$$

$$\leq \left( \frac{\epsilon^2 L^p R^p}{\log(1/\delta) + \log^2 d \log m} \right)^\ell + (\epsilon L^p R^p)^\ell \delta$$

$$\leq (\epsilon L^p R^p)^\ell \delta.$$

Therefore, taking expectation over $S$ while conditioned on $\mathcal{E}$, we have that $\mathbb{E}_{S,\sigma}[\Lambda^\ell | \mathcal{E}] \leq (\epsilon L^p R^p)^\ell \delta$. Rescaling $\epsilon = \epsilon/2$ completes the proof of (4), as desired. □

**Lemma A.3.** *Let* $\delta((\boldsymbol{\theta}_1, \boldsymbol{\theta}_2), (\boldsymbol{\theta}_1', \boldsymbol{\theta}_2'))$ *and* $\|\theta\|_J$ *be as defined in* (6) *and* (7), *respectively. It holds that*

$$\delta((\boldsymbol{\theta}_1, \boldsymbol{\theta}_2), (\boldsymbol{\theta}_1', \boldsymbol{\theta}_2')) \lesssim \begin{cases} d^{\frac{p-2}{4}} L^p R^{p-1} (\|\boldsymbol{\theta}_1 - \boldsymbol{\theta}_1'\|_J + \|\boldsymbol{\theta}_2 - \boldsymbol{\theta}_2'\|_J) & p \geq 2, \\ L^p R^{\frac{p}{2}} (\|\boldsymbol{\theta}_1 - \boldsymbol{\theta}_1'\|_J + \|\boldsymbol{\theta}_2 - \boldsymbol{\theta}_2'\|_J)^{\frac{p}{2}} & 1 \leq p \leq 2. \end{cases}$$

*Hence* $\operatorname{diam}(T)$, *the diameter of the subspace* $T$, *is at most* $O(d^{\max(\frac{p}{4},\frac{1}{2})} L^p R^p)$.

*Proof.* For $p \geq 2$, we have

$$\delta^2((\boldsymbol{\theta}_1, \boldsymbol{\theta}_2), (\boldsymbol{\theta}_1', \boldsymbol{\theta}_2')) \overset{(A)}{\leq} \sum_{k \in J} \lambda_k |\lambda_k^{-\frac{1}{p}} \boldsymbol{u}_k - \lambda_k^{-\frac{1}{p}} \boldsymbol{u}_k'|^2 (|\lambda_k^{-\frac{1}{p}} \boldsymbol{u}_k|^{p-1} + |\lambda_k^{-\frac{1}{p}} \boldsymbol{u}_k'|^{p-1})^2$$

$$\overset{(B)}{\leq} 2L^{2p} p \sum_{k \in J} \lambda_k \left( \frac{\left| \lambda_k^{-\frac{1}{p}} (\boldsymbol{a}_{i_k}^\top \boldsymbol{\theta}_1 - \boldsymbol{a}_{i_k}^\top \boldsymbol{\theta}_1') \right| + \left| \lambda_k^{-\frac{1}{p}} (\boldsymbol{a}_{i_k}^\top \boldsymbol{\theta}_2 - \boldsymbol{a}_{i_k}^\top \boldsymbol{\theta}_2') \right|}{(mp_{i_k})^{\frac{1}{p}}} \right)^2$$

$$\cdot \left( |\lambda_k^{-\frac{1}{p}} \boldsymbol{v}_k|^{2p-2} + |\lambda_k^{-\frac{1}{p}} \boldsymbol{v}_k'|^{2p-2} \right)$$

$$\overset{(C)}{\leq} 2^{p-1} p d^{\frac{p-2}{2}} L^{2p} R^{p-2} \sum_{k \in J} \lambda_k \left( \frac{\left| \lambda_k^{-\frac{1}{p}} (\boldsymbol{a}_{i_k}^\top \boldsymbol{\theta}_1 - \boldsymbol{a}_{i_k}^\top \boldsymbol{\theta}_1') \right| + \left| \lambda_k^{-\frac{1}{p}} (\boldsymbol{a}_{i_k}^\top \boldsymbol{\theta}_2 - \boldsymbol{a}_{i_k}^\top \boldsymbol{\theta}_2') \right|}{(mp_{i_k})^{\frac{1}{p}}} \right)^2$$

$$\cdot \left( |\lambda_k^{-\frac{1}{p}} \boldsymbol{v}_k|^p + |\lambda_k^{-\frac{1}{p}} \boldsymbol{v}_k'|^p \right)$$

$$\overset{(D)}{\leq} 2^{p-1} p d^{\frac{p-2}{2}} L^{2p} R^{p-2} (\|\boldsymbol{\theta}_1 - \boldsymbol{\theta}_1'\|_J + \|\boldsymbol{\theta}_2 - \boldsymbol{\theta}_2'\|_J)^2 \sum_{k \in J} \lambda_k (|\lambda_k^{-\frac{1}{p}} \boldsymbol{v}_k|^p + |\lambda_k^{-\frac{1}{p}} \boldsymbol{v}_k'|^p)$$

$$\overset{(E)}{\leq} 2^{2p-1} p d^{\frac{p-2}{2}} L^{2p} R^{2p-2} (\|\boldsymbol{\theta}_1 - \boldsymbol{\theta}_1'\|_J + \|\boldsymbol{\theta}_2 - \boldsymbol{\theta}_2'\|_J)^2,$$

where the inequality (A) follows from $|a|^p - |b|^p \leq p(|a|^{p-1} + |b|^{p-1})|a - b|$, (B) follows from triangle inequality and $(a + b)^2 \leq 2(a^2 + b^2)$, (C) follows from (5) and (E) is obtained by $\|v\|_p \leq \|SA\boldsymbol{\theta_1}\|_p + \|SA\boldsymbol{\theta_2}\|_p \leq 2R$.

For $1 \leq p \leq 2$, we have

$$\delta^2((\boldsymbol{\theta}_1, \boldsymbol{\theta}_2), (\boldsymbol{\theta}_1', \boldsymbol{\theta}_2'))$$

$$\leq \sum_{k \in J} \lambda_k |\lambda_k^{-\frac{1}{p}} \boldsymbol{u}_k - \lambda_k^{-\frac{1}{p}} \boldsymbol{u}_k'|^2 (|\lambda_k^{-\frac{1}{p}} \boldsymbol{u}_k|^{p-1} + |\lambda_k^{-\frac{1}{p}} \boldsymbol{u}_k'|^{p-1})^2$$

$$\leq \max_{k \in J} |\lambda_k^{-\frac{1}{p}} \boldsymbol{u}_k - \lambda_k^{-\frac{1}{p}} \boldsymbol{u}_k'|^p \cdot \sum_{k \in J} \lambda_k |\lambda_k^{-\frac{1}{p}} \boldsymbol{u}_k - \lambda_k^{-\frac{1}{p}} \boldsymbol{u}_k'|^{2-p} (|\lambda_k^{-\frac{1}{p}} \boldsymbol{u}_k|^{2p-2} + |\lambda_k^{-\frac{1}{p}} \boldsymbol{u}_k'|^{2p-2})$$

$$\leq L^p (\|\boldsymbol{\theta}_1 - \boldsymbol{\theta}_1'\|_J + \|\boldsymbol{\theta}_2 - \boldsymbol{\theta}_2'\|_J)^p (\sum_{k \in J} \lambda_k |\lambda_k^{-\frac{1}{p}} \boldsymbol{u}_k - \lambda_k^{-\frac{1}{p}} \boldsymbol{u}_k'|^p)^{\frac{2-p}{p}}$$

$$\cdot \left[ (\sum_{k \in J} \lambda_k |\lambda_k^{-\frac{1}{p}} \boldsymbol{u}_k|^p)^{\frac{2p-2}{p}} + (\sum_{k \in J} \lambda_k |\lambda_k^{-\frac{1}{p}} \boldsymbol{u}_k'|^p)^{\frac{2p-2}{p}} \right]$$

$$\leq L^p(\|\boldsymbol{\theta}_1 - \boldsymbol{\theta}_1'\|_J + \|\boldsymbol{\theta}_2 - \boldsymbol{\theta}_2'\|_J)^p (\sum_{k \in J} \lambda_k |\lambda_k^{-\frac{1}{p}} \boldsymbol{u}_k|^p + \lambda_k |\lambda_k^{-\frac{1}{p}} \boldsymbol{u}_k'|^p)^{\frac{2-p}{p}}$$

$$\cdot \left[ (\sum_{k \in J} \lambda_k |\lambda_k^{-\frac{1}{p}} \boldsymbol{u}_k|^p)^{\frac{2p-2}{p}} + (\sum_{k \in J} \lambda_k |\lambda_k^{-\frac{1}{p}} \boldsymbol{u}_k'|^p)^{\frac{2p-2}{p}} \right]$$

$$\leq 2^p L^{2p} R^p (\|\boldsymbol{\theta}_1 - \boldsymbol{\theta}_1'\|_J + \|\boldsymbol{\theta}_2 - \boldsymbol{\theta}_2'\|_J)^p,$$

where we use Hölder's inequality $\|fg\|_1 \leq \|f\|_\alpha \|g\|_\beta$ with $\alpha = \frac{p}{2-p}$ and $\beta = \frac{p}{2p-2}$ in the third line.

For $p \geq 2$, the diameter of $T$ is upper bounded by

$$\max_{(\boldsymbol{\theta}_1, \boldsymbol{\theta}_2) \in T, (\boldsymbol{\theta}_1', \boldsymbol{\theta}_2') \in T} \delta((\boldsymbol{\theta}_1, \boldsymbol{\theta}_2), (\boldsymbol{\theta}_1', \boldsymbol{\theta}_2'))$$

$$\leq 2^{\frac{2p-1}{2}} p d^{\frac{p-2}{4}} L^p R^{p-1} (\|\boldsymbol{\theta}_1 - \boldsymbol{\theta}_1'\|_J + \|\boldsymbol{\theta}_2 - \boldsymbol{\theta}_2'\|_J)$$

$$\leq 2^{\frac{2p-1}{2}} p d^{\frac{p}{4}} L^p R^p,$$

where we use the fact that $\|\boldsymbol{\theta}_1 - \boldsymbol{\theta}_1'\|_J \leq d^{\frac{1}{2}} R$ from (5). For $1 \leq p \leq 2$, the diameter of $T$ is upper bounded by $L^p R^{\frac{p}{2}} (\|\boldsymbol{\theta}_1 - \boldsymbol{\theta}_1'\|_J + \|\boldsymbol{\theta}_2 - \boldsymbol{\theta}_2'\|_J)^{\frac{p}{2}} \leq d^{\frac{1}{2}} L^p R^p$ where we obtain $\|\boldsymbol{\theta}_1 - \boldsymbol{\theta}_1'\|_J \leq d^{\frac{1}{p}} R$ from (5). $\qquad \square$

**Lemma A.4.** *Suppose that $A \in \mathbb{R}^{n \times d}$ and $t_1, \ldots, t_n \in \mathbb{R}$ such that $t_i \geq \beta w_i(A)$ for all $i$. Let $m = \sum_i t_i$ and $S \in \mathbb{R}^{m \times n}$ be a reweighted sampling matrix of with row sampling probabilities $p_1, \ldots, p_n$, where $p_i = \frac{t_i}{m}$. If $\beta \geq \epsilon^{-2} \log \frac{d}{\delta}$, then the $\ell_p$ Lewis weights of $SA$ are upper bounded by $\frac{2}{\beta}$ with probability at least $1 - \delta$.*

*Proof.* Let $\boldsymbol{a}_i \in \mathbb{R}^{d \times 1}$ be the $i$-th row of $A$. Without loss of generality, suppose $A^\top W^{1-\frac{p}{2}} A = I_d$. Hence, the Lewis weights of $A$ are $w_i^{\frac{2}{p}} = \boldsymbol{a}_i^\top (A^\top W^{1-\frac{p}{2}} A)^{-1} \boldsymbol{a}_i = \boldsymbol{a}_i^\top \boldsymbol{a}_i = \|\boldsymbol{a}_i\|_2^2$. We claim that

$$(1 - \epsilon) I_d \preceq \sum_{k=1}^m \frac{\boldsymbol{a}_{i_k} \boldsymbol{a}_{i_k}^\top}{m p_{i_k}} w_{i_k}^{1-\frac{2}{p}} \preceq (1 + \epsilon) I_d$$

holds with probability at least $1 - \delta$. Let $X_k = \frac{\boldsymbol{a}_{i_k} \boldsymbol{a}_{i_k}^\top}{p_{i_k}} w_{i_k}^{1-\frac{2}{p}}$ and then we have $\mathbb{E} X_k = I_d$. First, we have $\mathbb{E} X_k = I_d$ and $\|X_k - I_d\|_2 \leq 1 + \frac{\|\boldsymbol{a}_{i_k}\|_2^2}{w_{i_k}/d} w_{i_k}^{1-\frac{2}{p}} = 1 + \frac{m}{\beta}$. Besides, we have that

$$\|\mathbb{E}(X_k - I_d)\|_2^2 = \|\mathbb{E}(X_k - I_d)^\top (X_k - I_d)\|_2$$

$$= \|\mathbb{E} X_k^\top X_k - I_d\|_2$$

$$= \left\| \frac{w_{i_k}}{p_{i_k}} \cdot \mathbb{E} \frac{\boldsymbol{a}_{i_k} \boldsymbol{a}_{i_k}^\top w_{i_k}^{1-\frac{2}{p}}}{p_{i_k}} - I_d \right\|_2$$

$$= \left\| \frac{w_{i_k}}{p_{i_k}} \sum_{i=1}^n \boldsymbol{a}_i \boldsymbol{a}_i^\top w_{i_k}^{1-2/p} + I_d \right\|_2$$

$$\leq 1 + \frac{m}{\beta}.$$

By matrix Chernoff bound, it follows that

$$\Pr \left\{ \left\| \frac{1}{m} \sum_{k=1}^m (X_k - I_d) \right\|_2 \geq \epsilon \right\} \leq 2d \exp \left( \frac{-m \epsilon^2}{1 + d + (1 + d) \cdot \epsilon/3} \right)$$

$$\leq 2d \exp \left( -\beta \epsilon^2 \right)$$

Setting $\beta = \Theta(\frac{d}{\epsilon^2} \log \frac{d}{\delta})$ guarantees the failure probability to be at most $\delta$, proving the claim. Therefore, we have that

$$(1 - \epsilon) \left( \frac{d}{m} \right)^{1-\frac{2}{p}} I_d \preceq \left[ \sum_{k=1}^m \frac{\boldsymbol{a}_{i_k}}{(m p_{i_k})^{\frac{1}{p}}} \left( \frac{w_{i_k}}{d p_{i_k}} \right)^{1-\frac{2}{p}} \frac{\boldsymbol{a}_{i_k}^\top}{(m p_{i_k})^{\frac{1}{p}}} \right]^{-1} \preceq (1 + 2\epsilon) \left( \frac{d}{m} \right)^{1-\frac{2}{p}} I_d$$

holds with probability at least $1 - \delta$. Hence, it follows that

$$\frac{\boldsymbol{a}_i^\top}{(mp_i)^{1/p}} \cdot \left[ \sum_{k=1}^m \frac{\boldsymbol{a}_{i_k}}{(mp_{i_k})^{\frac{1}{p}}} \left( \frac{dp_{i_k}}{w_{i_k}} \right)^{\frac{2}{p}-1} \frac{\boldsymbol{a}_{i_k}^\top}{(mp_{i_k})^{\frac{1}{p}}} \right]^{-1} \cdot \frac{\boldsymbol{a}_i}{(mp_i)^{1/p}} \le (1+2\epsilon) \frac{d}{m} \left( \frac{w_{i_k}}{dp_{i_k}} \right)^{2/p}.$$

Applying (Chen et al., 2022, Lemma A.2) and setting $\epsilon = \frac{1}{2}$ gives that $w_i(SA) \le 2 \frac{d}{m} \frac{w_i}{dp_i} \le \frac{2}{\beta}$. $\quad\square$

**Lemma A.5.** *Let $p > 0$ and $p \neq 1$. Suppose that $A \in \mathbb{R}^{n \times d}$ and $w_i(A) \le 1/\beta$ for all $i$ and $\beta > 1$. Let $\Lambda = \max_{x:\|Ax\|_p \le 1} |\sum_{i=1}^n \sigma_i |(Ax)_i|^p|$, where $\sigma_1, \ldots, \sigma_n$ are independent Rademacher variables. Then the following tail bound holds:*

$$\Pr\left\{ \Lambda \ge \left[ C \frac{d^{\max\{\frac{p}{2}-1,0\}}}{\beta} \right]^{\frac{1}{2}} \left[ \log^2 d \log n + z \right] \right\} \le 2\exp(-z^2).$$

*Proof.* The tail bound is proven by Dudley's integral tail bound

$$\Pr\left\{ \sup_{t \in T} X_t \gtrsim \int_0^\infty \sqrt{\ln N(T, d, \epsilon)} d\epsilon + z \cdot \mathrm{diam}(T) \right\} \le 2\exp(-z^2),$$

where $N(T, d, \epsilon)$ is the $\epsilon$-covering number of $T$ and $\mathrm{diam}(T)$ is the diameter of the space $T$. In our setting, $T$ is the subspace $\{y = Ax : x \in \mathbb{R}^d\}$. From (Ledoux & Talagrand, 1991, Equation (15.17) and (15.18)), the diameter is bounded by $d^{\max(\frac{p}{4}, \frac{1}{2})}$. By Dudley's integral, we have $\mathbb{E}_\sigma \Lambda \lesssim \int_0^\infty \sqrt{\ln N(T, d, \epsilon)} d\epsilon$. The upper bound of the integral was proven in (Ledoux & Talagrand, 1991, Theorem 15.13), assuming that $w_i(A) \le d/n$. The same proof can go through when the upper bound of $w_i(A) \le 1/\beta$, with (Ledoux & Talagrand, 1991, (15.17)) replaced with

$$\mathbb{E}\Lambda \le \frac{3d^{\max(\frac{p}{2}, 1)}}{2n} + \left( \frac{2}{d\beta} \right)^{\frac{1}{2}} \Xi,$$

where

$$\Xi = \mathbb{E}_{\sigma_i} \sup_{x: \left\| W^{-\frac{1}{p}} Ax \right\|_p \le 1} \left| \sum_{i \in J} (\frac{w_i}{d})^{\frac{1}{2}} \sigma_i |x_i|^p \right|$$

In the proof of Ledoux & Talagrand (1991), $\lambda_i$ is our $\frac{w_i}{d}$, and the factor $\frac{1}{M}$ is replaced with $\frac{1}{d\beta}$ due to the change of Lewis weights' upper bound from $\frac{n}{M}$ to $\frac{1}{\beta}$.

The main difficulty is to upper bound $\Xi$, which is again done by using Dudley's integral. The argument to upper bound the integral in (Ledoux & Talagrand, 1991, Theorem 15.13) still goes through when the upper bound of Lewis weights is changed, yielding that $\Xi \le Cd^{\max(\frac{p}{4}, \frac{1}{2})} \log d \sqrt{\log n}$. Combining the diameter of $T$ and the inequality for $\mathbb{E}\Lambda$ gives us the result. $\quad\square$

## B  SUBSPACE EMBEDDING

We note that there are mainly two kinds of $\ell_p$ Lewis weight sampling. The first kind is to retain or discard each row independently. Specifically, the $i$-th row of $A$ is retained with probability $p_i$ and discarded with probability $1 - p_i$. The resulting sampled matrix $SA$ has a random number of rows. The second kind has a fixed, prescribed number $m$ of sampled rows. Each sample is i.i.d. chosen to be the $i$-th row of $A$ with probability $t_i/m$, where $t_1, \ldots, t_n$ are weights satisfying that $\sum_i t_i = m$. We use sampling of the second kind (recall Definition 3.3) in our algorithm. However, our main result (Theorem 3.5) still works for the first kind of sampling matrices, see Appendix C for details.

In this section, we give the sample complexity for $\ell_p$ subspace embedding with distortion $1 + \epsilon$ for $p > 2$, using both kinds of sampling schemes.

For $p > 2$, Woodruff & Yasuda (2023) consider the first kind of sampling and give a sample complexity of $O(\frac{d^{\frac{p}{2}}}{\epsilon^2} (\log^2 d \log n + \log \frac{1}{\delta}))$ for $\ell_p$-subspace-embeddings. This is the first result for $p > 2$ that has an $\epsilon^{-2}$ dependence, as the only prior result was $O(\epsilon^{-5} d^{p/2} \log d)$ with an $\epsilon^{-5}$ dependence (Bourgain et al., 1989). Still, based on the result of Bourgain et al. (1989), we can improve

the analysis of Woodruff & Yasuda (2023) and remove the undesired $\log n$ factor in their sample complexity. We have the following theorem.

**Theorem B.1.** *Let $A \in \mathbb{R}^{n \times d}$, $2 < p < \infty$ and $0 < \epsilon, \delta < 1$. Let $p_i = \min\{\beta w_i, 1\}$ where $w_i$ is $\ell_p$ Lewis weight of $a_i$ for $A$ and $\beta = \Omega(\frac{d^{\frac{p}{2}-1}}{\epsilon^2}(\log d + \log \frac{1}{\delta}))$ be the oversampling parameter. Let $S \in \mathbb{R}^{n \times n}$ be the reweighted sampling matrix in which the $i$-th row*

$$S_i = \begin{cases} \frac{1}{(p_i)^{1/p}} e_i^\top, & \text{with prob. } p_i \\ 0, & \text{with prob. } 1 - p_i. \end{cases}$$

*With probability at least $1 - \delta$, $S$ has $m = \Omega(\frac{d^{\frac{p}{2}}}{\epsilon^2}(\log^2 d \log \frac{d}{\epsilon} + \log \frac{1}{\delta}))$ nonzero rows and $\|SAx\|_p^p = (1 \pm \epsilon)\|Ax\|_p^p$.*

We only sketch the changes in the proof of Woodruff & Yasuda (2023). First, in the sampling we do not use $\gamma$-one-sided Lewis weights but the exact Lewis weights of $A$. True Lewis weights do not affect the symmetrization trick. After the symmetrization step, we remove the part of flattening matrix $A$ in their proof. Instead, we claim: (1) By Theorem 7.3 in (Bourgain et al., 1989), $S$ is a $\frac{1}{2}$-subspace embedding matrix of $A$. (2) By Lemma A.3 of (Chen et al., 2022), Lewis weights of $SA$ are uniformly upper bounded by $\frac{2}{\beta}$. Conditioned on (1) and (2), it suffices to prove

$$\mathbb{E}_{S,\sigma} \max_{x: \|SAx\|_p \leq 1} \left| \sum_{k=1}^{m} \sigma_k |(SA)_k x|^p \right|^\ell \leq \epsilon^\ell.$$

This $\ell$-th moment upper bound can be derived in the same fashion as the end of the proof of Lemma A.2. Then applying the Markov inequality gives us $\|SAx\|_p^p = (1 \pm \epsilon)\|Ax\|_p^p$ with probability at least $1 - \delta$.

The next theorem gives the sample complexity for $\ell_p$ subspace embedding for $p > 2$ in which samplings are i.i.d. and the probability of every row $a_i$ being sampled is $w_i/d$.

**Theorem B.2.** *Let $A \in \mathbb{R}^{n \times d}$, $2 < p < \infty$ and $0 < \epsilon, \delta < 1$. Suppose that the $\ell_p$ Lewis weights of $A$ are $w_1, \ldots, w_n$. Let $p_i = w_i/d$ and $S \in \mathbb{R}^{m \times d}$ be a reweighted sampling matrix whose $i$-th row $S_i = \frac{1}{(mp_i)^{1/p}} e_j^\top$ with probability $p_j$. Set $m = \Omega(\frac{d^{\frac{p}{2}}}{\epsilon^2}(\log^2 d \log \frac{d}{\epsilon} + \log \frac{1}{\delta}))$, then with probability at least $1 - \delta$, we have $\|SAx\|_p^p = (1 \pm \epsilon)\|Ax\|_p^p$.*

We only highlight the necessary changes in the proof of Theorem B.2.

- The symmetrization step goes through in the same fashion as the long chain of inequalities in the proof of Lemma A.2.
- By Theorem 7.3 of (Bourgain et al., 1989), $S$ is a $\frac{1}{2}$-subspace embedding matrix of $A$.
- By Lemma A.4, Lewis weights of $SA$ are uniformly upper bounded by $\frac{2}{\beta}$. The left steps are the same as the changes mentioned for Theorem B.1.

## C  RESULT FOR THE OTHER SAMPLING METHOD

In this section, we prove that our main result Theorem 3.5 still holds if the reweighted sampling matrix $S$ is defined to be of the first kind:

$$S_i = \begin{cases} \frac{1}{(p_i)^{1/p}} e_i^\top, & \text{with prob. } p_i \\ 0, & \text{with prob. } 1 - p_i, \end{cases}$$

where $p_i = \beta w_i$ and $\beta = \Omega(\frac{d^{\frac{p}{2}-1}}{\epsilon^2}(\log^2 d \log \frac{d}{\epsilon} + \log \frac{1}{\delta}))$. Accordingly, Lines 3–8 of Algorithm 1 are changed to the following lines.

1: **for** $i = 1, 2, \ldots, n$ **do**
2:     **if** $a_i$ is sampled with probability $p_i = \beta w_i$ **then**
3:         $S_{i,i} = p_i^{-1/p}$ and query the label of $a_i$

Compared to the proof of Theorem 3.5, the following modifications are needed: (1) By Theorem B.1, $S$ is a $1/2$-subspace embedding matrix of $A$. (2) To show that Lemma A.2 holds, we observe that by Lemma A.3 of (Chen et al., 2022), Lewis weights of $SA$ are uniformly upper bounded by $\frac{2}{\beta}$.

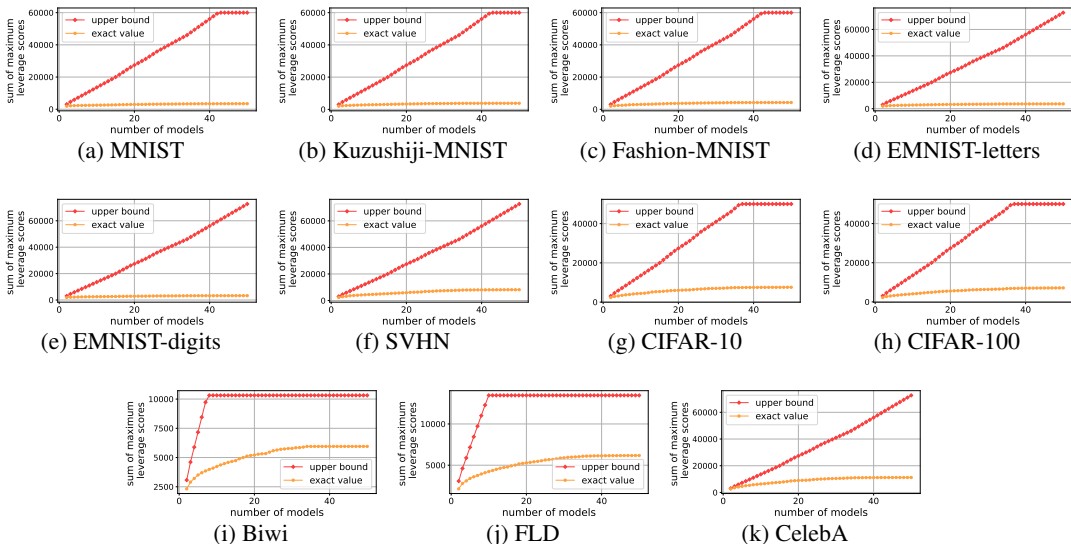

Figure 4: The trends of the sum of the maximum Lewis weights with $p = 2$ among multiple representations as the number of deep models increases.

## D  DETAILS OF EMPIRICAL OBSERVATION

To address the challenges of distinct representations from multiple deep models, one solution is to sample the unlabeled instances by their maximum Lewis weight among different representations. Recall that the sample size is proportional to the sum of the sampling probabilities, this strategy will not save the number of queries if the sum of the maximum Lewis weights is $k$ times larger than that of a single regression problem. Therefore, we would like first to examine the sum of maximum Lewis weights across representations as it will determine the potential query savings. In the following empirical studies, we mainly consider the case of $p = 2$ (i.e., squared loss), where the Lewis weight becomes exactly the leverage score.

**Empirical Settings.** We conduct experiments on 11 datasets, the details are summarized in Table 3. For each dataset, we randomly sample 3000 instances to train the models with 20 epochs, then extract features for the whole dataset and calculate their leverage scores. We use the squared loss in model training and keep all other settings the same as the OFA project. We employ 50 distinct network architectures as the target models. These architectures are published by a recent NAS method OFA (Cai et al., 2019) for accommodating diverse resource-constraint devices, ranging from NVIDIA Tesla V100 GPU to mobile devices. To demonstrate the distinctness of these models, we report the initial performances of the models on the classification datasets in Table 2. It can be observed that the model performances are significantly diverse. It aligns well with our problem

Table 2: Summary of the initial performances of 50 deep models on the classification datasets. We report the mean accuracy, standard deviation of the accuracies, maximum and minimum accuracy in the table.

| Dataset | mean accuracy | std. deviation | maximum | minimum |
|---|---|---|---|---|
| MNIST | 94.79 | 4.39 | 97.61 | 72.99 |
| F.MNIST | 81.76 | 3.03 | 85.09 | 67.32 |
| K.MNIST | 74.79 | 7.94 | 85.61 | 51.31 |
| CIFAR-10 | 48.54 | 3.36 | 53.01 | 36.92 |
| CIFAR-100 | 11.34 | 2.14 | 16.85 | 7.63 |
| SVHN | 45.53 | 7.62 | 60.98 | 31.88 |
| EMNIST-l. | 70.04 | 14.21 | 83.23 | 25.84 |
| EMNIST-d. | 95.12 | 3.64 | 97.76 | 76.13 |

Table 3: The specifications of the datasets used in the experiments.

| Dataset | #Training | #Testing | #Label | Task |
|---|---|---|---|---|
| MNIST (LeCun et al., 1998) | 60,000 | 10,000 | 10 | Classification |
| Fashion-MNIST (Xiao et al., 2017) | 60,000 | 10,000 | 10 | Classification |
| Kuzushiji-MNIST (Clanuwat et al., 2018) | 60,000 | 10,000 | 10 | Classification |
| SVHN (Netzer et al., 2011) | 73,257 | 26,032 | 10 | Classification |
| EMNIST-digits Cohen et al. (2017) | 240,000 | 40,000 | 10 | Classification |
| EMNIST-letters (Cohen et al., 2017) | 88,800 | 14,800 | 26 | Classification |
| CIFAR-10 (Krizhevsky, 2009) | 50,000 | 10,000 | 10 | Classification |
| CIFAR-100 Krizhevsky (2009) | 50,000 | 10,000 | 100 | Classification |
| Biwi (Fanelli et al., 2013) | 10,317 | 5,361 | 2 | Regression |
| FLD (Sun et al., 2013) | 13,466 | 249 | 10 | Regression |
| CelebA (Liu et al., 2015) | 162,770 | 19,962 | 10 | Regression |

setting. Figure 1 shows the theoretical upper bound and the exact values of the sum of the maximum leverage score of each instance across different representations.

**Results.** We first plot the upper bound of the sum of the maximum leverage scores across multiple representations as the number of models increases. For matrices $A^1, \ldots, A^k$ of $n$ rows, it clearly holds that $\sum_i \max_j w_i(A^j) \leq \min\{\sum_i \sum_j w_i(A^j), \sum_i 1\} \leq \min\{\sum_i \text{rank}(A^j), n\}$. This upper bound is plotted in red color, which grows almost linearly until it reaches the number of instances $n$.

We examine the exact values of the sum of maximum leverage scores across multiple representations. All figures show that the exact sum grows very slowly as the number of models increases. This suggests highly consistent discrimination power of most instances across different representations, as the leverage score when $p = 2$ is exactly the leverage score, which measures how hard an instance can be linearly represented by others. Therefore, a small number of discriminating examples suffices to effectively train multiple models. leverage scores also provide a possible direction to interpret the behavior of deep representation learning, as prior works have not discovered any simple form of correlation among the diverse representations obtained by different model architectures (Kornblith et al., 2019; Nguyen et al., 2020).

# E  DETAILED EXPERIMENTAL SETTINGS AND ADDITIONAL RESULTS

## E.1  DETAILED SETTINGS

All experiments are run on two GPU servers, each equipped with four GeForce RTX 3090 graphic cards, an Intel Xeon Gold 5317 CPU and 128 GB Memory. The details of the datasets are presented in Table 3. The configurations of 50 deep models are specified in Table 4. Note that the network dentition and pre-trained weights of each configuration can be downloaded from the GitHub page of the OFA project (Cai et al., 2019).

We follow the training settings of (Tang & Huang, 2022) in our experiment of the vanilla deep learning scenario. Specifically, at each query iteration, each target model will be initialized with the pre-trained weights on ImageNet and trained for 20 epochs on the labeled dataset with batch size 32. SGD optimizer is employed with learning rate $1.5 \times 10^{-3}$, momentum coefficient 0.9 and weight decay factor $3 \times 10^{-5}$. A dropout rate of 0.1 is used in the training process.

In our experiments of the fine-tuning scenario, we train a linear prediction layer with ReLU activation function using mean squared loss. The training specifications are introduced as follows. The SGD optimizer is employed with a learning rate of $10^{-3}$ and a weight decay coefficient of $10^{-1}$. The layer is trained for 30 epochs with training batch size 128. We set the random seed to 0 for reproducibility. Please refer to the submitted source code to reproduce our results.

The implementations of each compared method are introduced as follows. We use the code in (Tang & Huang, 2022) to implement DIAM, Coreset and Entropy methods. Specifically, DIAM first obtains the predictions of the unlabeled instances using the models in the latter half of training epochs of each target network. Then, it selects the batch of instances that multiple models have inconsistent

Table 4: The names of the model specifications used in the experiments. The network dentition and pre-trained weights of each configuration can be downloaded from the GitHub page of the OFA project (Cai et al., 2019).

| | |
|---|---|
| flops@595M_top1@80.0_finetune@75 | flops@482M_top1@79.6_finetune@75 |
| flops@389M_top1@79.1_finetune@75 | LG-G8_lat@24ms_top1@76.4_finetune@25 |
| LG-G8_lat@16ms_top1@74.7_finetune@25 | LG-G8_lat@11ms_top1@73.0_finetune@25 |
| LG-G8_lat@8ms_top1@71.1_finetune@25 | s7edge_lat@88ms_top1@76.3_finetune@25 |
| s7edge_lat@58ms_top1@74.7_finetune@25 | s7edge_lat@41ms_top1@73.1_finetune@25 |
| s7edge_lat@29ms_top1@70.5_finetune@25 | note8_lat@65ms_top1@76.1_finetune@25 |
| note8_lat@49ms_top1@74.9_finetune@25 | note8_lat@31ms_top1@72.8_finetune@25 |
| note8_lat@22ms_top1@70.4_finetune@25 | note10_lat@64ms_top1@80.2_finetune@75 |
| note10_lat@50ms_top1@79.7_finetune@75 | note10_lat@41ms_top1@79.3_finetune@75 |
| note10_lat@30ms_top1@78.4_finetune@75 | note10_lat@22ms_top1@76.6_finetune@25 |
| note10_lat@16ms_top1@75.5_finetune@25 | note10_lat@11ms_top1@73.6_finetune@25 |
| note10_lat@8ms_top1@71.4_finetune@25 | pixel1_lat@143ms_top1@80.1_finetune@75 |
| pixel1_lat@132ms_top1@79.8_finetune@75 | pixel1_lat@79ms_top1@78.7_finetune@75 |
| pixel1_lat@58ms_top1@76.9_finetune@75 | pixel1_lat@40ms_top1@74.9_finetune@25 |
| pixel1_lat@28ms_top1@73.3_finetune@25 | pixel1_lat@20ms_top1@71.4_finetune@25 |
| pixel2_lat@62ms_top1@75.8_finetune@25 | pixel2_lat@50ms_top1@74.7_finetune@25 |
| pixel2_lat@35ms_top1@73.4_finetune@25 | pixel2_lat@25ms_top1@71.5_finetune@25 |
| 1080ti_gpu64@27ms_top1@76.4_finetune@25 | 1080ti_gpu64@22ms_top1@75.3_finetune@25 |
| 1080ti_gpu64@15ms_top1@73.8_finetune@25 | 1080ti_gpu64@12ms_top1@72.6_finetune@25 |
| v100_gpu64@11ms_top1@76.1_finetune@25 | v100_gpu64@9ms_top1@75.3_finetune@25 |
| v100_gpu64@6ms_top1@73.0_finetune@25 | v100_gpu64@5ms_top1@71.6_finetune@25 |
| tx2_gpu16@96ms_top1@75.8_finetune@25 | tx2_gpu16@80ms_top1@75.4_finetune@25 |
| tx2_gpu16@47ms_top1@72.9_finetune@25 | tx2_gpu16@35ms_top1@70.3_finetune@25 |
| cpu_lat@17ms_top1@75.7_finetune@25 | cpu_lat@15ms_top1@74.6_finetune@25 |
| cpu_lat@11ms_top1@72.0_finetune@25 | cpu_lat@10ms_top1@71.1_finetune@25 |

predictions. Coreset selects data points based on the representation of the pre-trained super-net in OFA. Entropy calculates the entropy scores of unlabeled instances based on the predictions of each target model. Subsequently, it selects the instances with the highest mean entropy scores across multiple model predictions.

## E.2 ADDITIONAL RESULTS

### E.2.1 STUDY ON MEAN PERCENTAGE OF COVERED INSTANCES

We further examine how many instances with high leverage scores under the representation of a single model can be covered by the maximum leverage score sampling. The statistics are calculated as follows: We first get the intersection between the sets of instances that have the top $t\%$ highest leverage score under the representation of model $j$ (denoted by $I_j^t$) and top $t\%$ highest Maximum leverage score (denoted by $I^t$). Then, we divide the cardinality of this subset by the number of $t\%$ unlabeled instances. Finally, we calculate this value for each $j \in [k]$ and compute the average to obtain the mean percentage of covered instances, i.e.,

$$\kappa(t) = \frac{1}{k} \sum_{j=1}^{k} \frac{|I_j^t \cap I^t|}{|I^t|}.$$

We report the mean percentage of covered instances of 50 deep models in Figure 5. It can be observed that $\kappa(10)$ is about 30% on most datasets (except for Biwi), that is, $I^{10}$ covers on average about 30% of the instances with high leverage scores of each representation for most datasets. For all datasets, as $t$ increases, $\kappa(t)$ increases rapidly. These phenomena suggest that sampling a modest number of instances by maximum leverage scores can effectively train multiple deep models, as there are a significant fraction of instances with high leverage scores shared across different models.

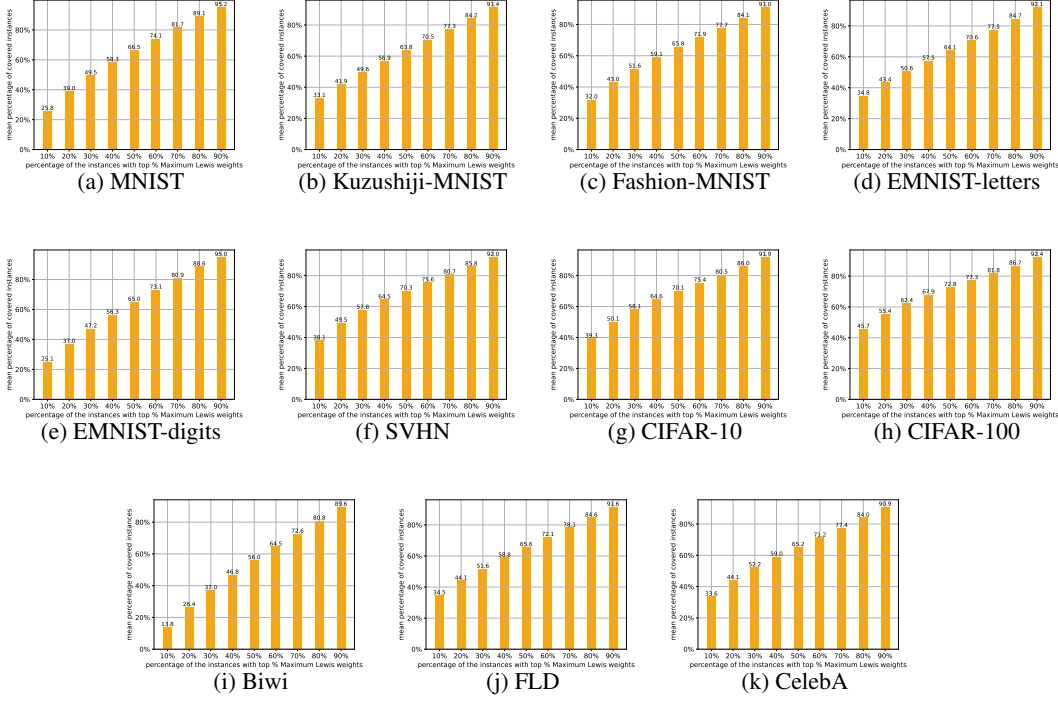

Figure 5: The mean percentage of shared data between instances having the highest maximum leverage score and those having the highest leverage score under the representation of a specific deep model.

### E.2.2 STUDY ON THE CLASS IMBALANCE RATIO

Another metric of interest for AL classification algorithms is the class imbalance ratio, which is defined as $\max_c \sum_{i \in [n_l]} \mathbb{I}\{y_i = c\} / \min_c \sum_{i \in [n_l]} \mathbb{I}\{y_i = c\}$, where $\mathbb{I}$ is the indicator function. Some active query strategies may cause severe class imbalance, rendering them hardly generalizable to other target models and learning tasks. This issue becomes more significant for multiple target models. In this experiment, we examine the class imbalance ratios of different AL methods in the classification tasks. Specifically, we compare the ratios after the third AL iteration, where a total of 12000 labeled instances are present, including the initially labeled set. The results are reported in Table 5.

We can observe that our proposed method consistently produces a balanced labeled dataset. Its class imbalanced ratio is very close to that of the Random sampling. On the other hand, Coreset suffers from the class imbalance, suggesting that the training instances with different classes exhibit diverse intra-class distances under deep representation. This implies that the instances sampled by Coreset may have a large distribution gap with the dataset. Entropy obtains a similar class imbalance ratio to that of Random, which might appear counter-intuitive, given that Entropy prefers the instances near the decision boundary and such instances are typically less likely to be class-balanced. A possible reason is that the mean entropy scores of the unlabeled instances may have a small standard deviation, potentially diminishing the advantages of using Entropy for identifying the most informative instances, in the setting of multiple target models.

Table 5: The class imbalance ratio of different query strategies.

|        | MNIST | F.MNIST | K.MNIST | SVHN  | CIF.10 | CIF.100 | EMN.l. | EMN.d. |
|--------|-------|---------|---------|-------|--------|---------|--------|--------|
| DIAM   | 2.007 | 3.250   | 1.472   | 2.121 | 1.525  | 4.390   | 2.216  | 2.811  |
| QBC    | 1.867 | 6.721   | 1.987   | 4.474 | 5.212  | 13.607  | 9.178  | 10.664 |
| Coreset| 3.561 | 3.708   | 2.116   | 2.330 | 1.871  | 6.000   | 5.375  | 4.491  |
| Random | 1.262 | 1.091   | 1.067   | 3.008 | 1.062  | 1.511   | 1.092  | 1.222  |
| Entropy| 1.331 | 1.077   | 1.091   | 3.052 | 1.087  | 1.439   | 1.088  | 1.166  |
| Our    | 1.217 | 1.081   | 1.050   | 2.993 | 1.098  | 1.440   | 1.109  | 1.209  |

Table 6: Running time (hours) of the methods in regression benchmarks. The running time includes model training and data querying.

| Methods | Biwi  | FLD   | CelebA |
|---------|-------|-------|--------|
| Random  | 1.217 | 1.383 | 1.202  |
| Coreset | 1.483 | 1.883 | 3.667  |
| Our     | 1.551 | 1.850 | 5.817  |
| QBC     | 1.632 | 1.800 | 4.110  |

