# OpenReview forum: "One-shot Active Learning Based on Lewis Weight Sampling for Multiple Deep Models"
_ICLR.cc/2024/Conference — ICLR 2024 poster_

### Official Review · Reviewer_QZho · 2023-10-28

**Soundness:** 3 good
**Presentation:** 2 fair
**Contribution:** 2 fair
**Rating:** 5
**Confidence:** 3

**Summary:**

This study introduces a one-shot AL approach aimed at reducing the computational overhead caused by iterative model training. The strategy for sampling unlabeled instances leverages the maximum Lewis weights across various representations. The authors validate their method on eight classification and regression benchmarks, employing 50 deep learning models to demonstrate its efficiency.

**Strengths:**

- This work aims to present a more comprehensive framework, applicable to multiple models and multiple error norms.
- The paper is well-structured and easy to follow.
- The motivation behind the sampling strategy is easy to comprehend.

**Weaknesses:**

- If the results from the empirical study suggest that as the number of models increases, the sum of the Lewis weights grows very slowly, this may indeed imply that the information between different models might not be so "distinct". If I understand correctly, isn't there a certain gap between this and the motivation mentioned in the introduction?
- In the experiments, why are the baseline settings different for the classification task and the fine-tuning regression task, with the former being iterative and the latter one-shot?
- In the experiments, the running time of data querying and model training for different methods in classification benchmarks was presented. What about in the fine-tuning scenario?
- Why isn't there a one-shot method for multiple models in the baseline?

**Questions:**

- Can this method be generalized to situations where different models have differen norms in their training objective functions?

---

> ### Author Response · Authors · 2023-11-20
>
> > The slow growth rate of the maximum Lewis weight imply that the information between different models might not be so "distinct". If I understand correctly, isn't there a certain gap between this and the motivation mentioned in the introduction?
>
> Thanks for the comments. We argue that the target models do have significant distinctions. We demonstrate this from two aspects. First, we have reported the network specifications in the appendix. The detailed network architectures and MACs can be found at the project page of OFA [1] (https://github.com/mit-han-lab/once-for-all). We can see that their accuracy, MACs, number of parameters and the applicable devices are very different. Second, we report the initial performances of these deep models in our experiment in Table 2 in the revised paper.
> It can be observed from Table 2 that the model performances are significantly diverse.  Therefore, we hold the point that the target models have significant distinctions and the slow growth rate suggests highly consistent discrimination power of most instances across different deep representations.
>
> > Why are the baseline settings different for the classification task and the fine-tuning regression task, with the former being iterative and the latter one-shot?
>
> We clarify that the Coreset and Random baselines are actually one-shot methods in the vanilla deep learning scenarios. Most one-shot AL methods only depend on the features to select data. They can be employed on both classification and regression tasks, such as Coreset and Random. However, most AL for classification methods require repetitive model updates. Therefore, we implement all the compared methods in an iterative way in this scenario for better comparison. We have clarified this point in the revised paper. Thank you.
>
>
> > What is the running time of the compared methods in the fine-tuning scenario?
>
> In the fine-tuning scenario, all the compared methods conduct one-shot querying and linear prediction layers are trained with the same computational costs. As a result, the running time of the compared methods is comparable. That is why we did not report them in the paper. We have supplemented these results to the revised paper (Table 6).
>
> > Why isn't there a one-shot method for multiple models in the baseline?
>
> We clarify that the Coreset and Random baselines are actually one-shot methods in the vanilla deep learning scenarios. Most AL methods for classification need to acquire the model predictions of the unlabeled data and are thus less applicable for one-shot querying. We have made this clearer in the revised paper, thank you.
>
> > Can this method be generalized to situations where different models have different norms in their training objective functions?
>
> Yes, it can be generalized and our theorem will still hold. When different models use different values of $p$, we can calculate different $\ell_p$ Lewis weights and still take the maximum across the models. Then, we use different reweights $(mp_j)^{-1/p}$ for different models according to their corresponding value of $p$ to train the linear prediction heads.
>
> [1] Han Cai, Chuang Gan, Tianzhe Wang, Zhekai Zhang, and Song Han. Once-for-all: Train
> one network and specialize it for efficient deployment. In Proceedings of the 7th International Conference on Learning Representations, 2019.

---

> > ### Comment · Reviewer_QZho · 2023-11-21
> > **Response**
> >
> > Thank you for your response regarding the distinct characteristics of the target models, as outlined in the appendix and in Table 2. However, I'm still curious about a specific aspect:
> >
> > Considering the distinct embedding spaces of these different models, how do you account for the observed uniformity in the performance of most instances across such diverse models?
> >
> > On the other hand, my understanding, based on the information in Table 1 and as suggested by Figure 1, is that despite the differences in training, deployment, and performance, there seems to be a level of consistency in the embedding spaces of these models. This consistency might facilitate the use of a limited number of samples for effective simultaneous optimization across various models.
> >
> > If my understanding is correct, perhaps you could further clarify or expand on the motivation. If not, I am interested in understanding how these different models maintain such consistent performance in the same instances despite their varied embedding spaces.

---

> > > ### Author Response · Authors · 2023-11-21
> > >
> > > We believe that the embeddings extracted from 50 deep models have distinctions. One piece of evidence supporting this belief is the highly consistent performance between Entropy and Random methods, as shown in Figure 2 and Table 5. Otherwise, the Entropy method would have queried more discriminative instances.
> > >
> > > The reasons why the same set of limited data can effectively improve diverse models may come from two aspects. On the one hand, certain instances are informative for all models. Since the learning task for all models is the same, some instances may be crucial in minimizing the objective function. On the other hand, our method takes into account the needs of all models. It samples instances according to their maximum Lewis weights and reweights the queried instances to ensure that the sample norm $||SA\mathbf{x}||_2$ is an unbiased estimator of $||A\mathbf{x}||_2$ for all model embeddings, making it more effective than other AL algorithms. We believe this is also an advantage of the proposed method.

---

### Official Review · Reviewer_ecLZ · 2023-10-31

**Soundness:** 2 fair
**Presentation:** 3 good
**Contribution:** 2 fair
**Rating:** 5
**Confidence:** 3

**Summary:**

The paper discusses the use of a one-shot method for active learning to save on the running time of retraining models. The base of the algorithmic development is a solution to the p-th regression problem involving the sampling matrix . The authors also show improvement on prior work for p=2 from O(d^2/e\(\epsilon^4) to O(d/(\epsilon^4)). Empirical results are presented on comparison with other AL algorithms, showing comparable results and reduced running time.

**Strengths:**

1.	Claims improvement on prior work in terms of sample complexity
2.	Reduced running time for the one-shot approach vs AL.
3.	An approach that tackles real problem of having multiple models

**Weaknesses:**

1.	Presentation is poorly motivated in many cases
2.	The empirical results cover a rather limited scope of data sets, most are MNIST types data sets and some curves don’t reflect SOTA (see questions below). The experimental setting is also counterintuitive for active learning scenarios.
3.	I am missing the connection between the theoretical & algorithmic construction and the empirical validation.
4.	Clarity:
a.	It isn’t clear what is the relation between the A matrix, the models and the data. Sometimes A is referred to as data and sometimes it is referred to as a model
b.	Why is the reweighting done by sqrt(mp_q)^-1? What is the motivation\intuition for that?

**Questions:**

1.	What is p in the in the experimental settings?
2.	Why are you starting with 3000 points for initialization? Isn’t that too much (e.g. for MNIST)? Does actual active learning start of with a random 3000 points? isn’t there too much redundancy in querying 3000 point on each query step?
3.	How come random performs better than other AL methods? E,g, Coreset in cifar 10? This does not correspond to the results reported in Senar et. al.
4.	What is L^2 in (1)? Is it the Lipschitz constant?
5.	Can you please show me where your claim about a reduced sample complexity for p=2 is proven? I’ve looked at the supplemental material and still haven’t seen it. If this is indeed a result it should also be addressed in your theorems, just as Gajjar et. al. do….

---

> ### Author Response · Authors · 2023-11-20
> **Part 1 of the response**
>
> > Presentation is poorly motivated in many cases.
>
> Thank you for pointing out this issue. We have improved the presentation of the paper, including using clearer notations to mitigate potential confusion, reorganizing preliminaries to make them easier to read, introducing the empirical settings in a more explicit way and others. Please see our revised paper.
>
> > The empirical evaluations are weak.
>
> We have used 3 additional datasets in our experiment, i.e., CIFAR-100, EMNIST-digits and CelebA, where CIFAR-100 and EMNIST-digits are classification benchmarks, CelebA is a regression benchmark. Specifically, we use the facial landmarks labels in CelebA to compare the performances of different AL methods.
> We follow all the empirical settings of other datasets in our experiment. The results are supplemented to Figure 2 and Figure 3 in the revised paper.
> We can observe that our method is consistently better than the other baselines in these datasets.
>
> > What’s the connection between the theoretical, algorithmic and empirical algorithms in the paper.
>
> Our theoretical results match exactly with our Algorithm 1. In Algorithm 1, lines 1 and 2 calculate the normalized maximum Lewis weight under multiple representations; Lines 3 to 8 sample the unlabeled data according to their maximum Lewis weights; Lines 9 to 11 calculate the reweights. These procedures correspond to the construction of the reweighted sampling matrix $S$ in our theorem. Finally, lines 12 to 15 train the linear prediction heads with the reweighted sampled data. This step corresponds to the learning of $\tilde{\mathbf{x}}$ in our theorem.
> In the experiment, there are two learning scenarios. In the fine-tuning scenario, our implementations remove the constraint $E$ in Line 15 in Algorithm 1 for better examination of its practicability. In the vanilla deep learning scenario, we use the default training scheme of deep models to replace Line 15 in Algorithm 1. We have supplemented more explanations in the revised paper.
>
> > It isn’t clear what is the relation between the A matrix, the models and the data.
>
> The matrix $A$ represents the feature matrix of the data. To mitigate the confusion, we have updated our notations. Specifically, we now use $\mathbf{\theta}$ to represent the linear model parameters, rather than $\mathbf{x}$.
>
> > Why is the reweighting done by $\sqrt{mp}^{-1}$? What is the motivation or intuition for that?
>
> The sampled data is reweighted by $\sqrt{mp_i}^{-1}$, where $p_i$ is the sampling probability of the $i$-th row of $A$. This is to ensure the sample norm $||SA\mathbf{x}||_2$ is an unbiased estimator of $||A\mathbf{x}||_2$, i.e., the expectation of $||SA\mathbf{x}||_2$ (over the randomness of $S$) equals exactly $||A\mathbf{x}||_2$. Then by concentration inequalities, we can show that $||SA\mathbf{x}||_2$ is close to its mean, i.e., $||A\mathbf{x}||_2$ with a large probability.
>
> > What is $p$ in the experimental settings?
>
> In our experiment, we use $p=2$. We mentioned this at the end of the empirical setting that our method uses leverage scores to select data and the model is trained with MSE loss. In Section 3.1, we state below Definition 3.1 that when $p=2$, Lewis weights are leverage scores.
> We have made this more explicit in the experiment section of our revised paper.

---

> ### Author Response · Authors · 2023-11-20
> **Part 2 of the response**
>
> > Why are you starting with 3000 points for initialization? Isn’t that too much (e.g. for MNIST)? Does actual active learning start of with a random 3000 points? isn’t there too much redundancy in querying 3000 point on each query step?
>
> In deep active learning, we believe a query batch size of around 3000 instances is generally accepted. For example, Sinha et al. [1] conduct experiments on CIFAR-10, CIFAR-100, Caltech-256, ImageNet, Cityscapes and BDD100K datasets. They use 10\% of the training set as initially labeled set and 5\% of the training dataset as the query batch size. Besides, many recent works [2][3] use a query batch size in the range of 2000 to 5000 instances in deep classification datasets, such as CIFAR-10, CIFAR-100, SVHN.
>
> Referring to these works, we unify the query batch size in the classification datasets as 3000 instances in our experiment.
> As these datasets have more than 60000 training instances (MNIST, CIFAR-10, CIFAR-100, SVHN, etc.). Given this, 3000 instances constitute less than 5\% of the training dataset.
>
> > Why random performs better than other AL methods? E,g, Coreset in CIFAR-10?
>
> The problem settings of Senar et al. [4] and our work are different. In our experiments, there are 50 distinct target networks to be learned. We follow the implementation of Tang and Huang [5] to adapt Coreset method to the multiple models setting. It solves the coreset problem based on the features extracted by the supernet. The selected instances may not be useful for other models, because the data representations are different. We believe this is the reason why Coreset is less effective than Random in our setting. We have supplemented more explanations of this phenomenon in the revised paper.
>
> > What is $L^2$ in (1)? Is it the Lipschitz constant?
>
> Yes, it is the Lipschitz constant of $f$. We state in an earlier paragraph that $f : \mathbb{R}\to \mathbb{R}$ is an $L$-Lipschitz function with $f(0) = 0$. This implies that $L$ is the Lipschitz constant of $f$. We have made this clearer in this revised paper.
>
> > Can you please show me where your claim about a reduced sample complexity for $p=2$ is proven?
>
> We prove Theorem 1.1. The proof is presented in the appendix A. As explained below Theorem 1.1, when the theorem is specialized to $p=2$ and to a single matrix, it implies a sampling complexity of $\tilde{O}(d/\epsilon^4)$. This is a factor of $d$ better than $\tilde{O}(d^2/\epsilon^4)$ in Gajjar et al. [6].
>
>
> [1] Samarth Sinha,. Variational adversarial active learning. In Proceedings of the IEEE/CVF International Conference on Computer Vision, pages 5972–5981, 2019.
>
> [2] Yonatan Geifman and Ran El-Yaniv. Deep active learning with a neural architecture
> search. In Advances in Neural Information Processing Systems, pages 5974–5984, 2019.
>
> [3] Gui Citovsky, Giulia DeSalvo, Claudio Gentile, Lazaros Karydas, Anand Rajagopalan,
> Afshin Rostamizadeh, and Sanjiv Kumar. Batch active learning at scale. In Advances in
> Neural Information Processing Systems, pages 11933–11944, 2021.
>
> [4] Ozan Sener and Silvio Savarese. Active learning for convolutional neural networks: A
> core-set approach. In Proceedings of the 6th International Conference on Learning Representations, 2018.
>
> [5] Ying-Peng Tang and Sheng-Jun Huang. Active learning for multiple target models. In Advances in Neural Information Processing Systems, 2022.
>
> [6]Aarshvi Gajjar, Christopher Musco, and Chinmay Hegde. Active learning for single neuron models
> with lipschitz non-linearities. In Procedings of the 26th International Conference on Artificial
> Intelligence and Statistics, pp. 4101–4113. PMLR, 2023.

---

> > ### Comment · Reviewer_ecLZ · 2023-11-22
> > **Thank you for your response, some of my concerns are addressed**
> >
> > Thank you for adding more datasets to your experiments. It reinforces your proposal.
> > Im still not completely convinced about the use of 3000 points as initialization, nevertheless some of the questions and concerns I have had have been answered.

---

### Official Review · Reviewer_pbqA · 2023-10-31

**Soundness:** 3 good
**Presentation:** 2 fair
**Contribution:** 3 good
**Rating:** 6
**Confidence:** 3

**Summary:**

This paper explores the domain of one-shot active learning for multiple target models. The authors introduce a novel active learning approach that leverages Lewis weights across representations from various target models. The authors provide both sample complexity analysis and empirical results. The empirical results indicate that the proposed method achieves performance comparable to those of iterative active learning methods.

**Strengths:**

a. This paper delves into a compelling and practical setting, one-shot active learning for multiple models, which holds significance in real-world scenarios.

b. The proposed approach, employing maximum Lewis weights, is both innovative and well-grounded. The sample complexity analysis presented in the paper is sound, with the derived bound improving upon prior results by a factor of d.

c. The paper's empirical results are promising. The authors compare their proposed approach with several iterative baselines, including the state-of-the-art method DIAM, demonstrating a substantial improvement in efficiency without sacrificing performance.

**Weaknesses:**

a. While the empirical results are promising, the datasets used in the experiments are relatively small for deep learning models. It would be valuable to assess the method's performance on larger datasets, such as CelebA, to evaluate its scalability and generalizability.

b. A more comprehensive evaluation could include an assessment of how the proposed approach compares with state-of-the-art general active learning methods in the multi-model setting, e.g., BADGE[1].

c. The paper's writing, particularly in Section 3.1, may benefit from further refinement. The section lists several definitions but offers limited explanations and intuitions, which may challenge readers who are unfamiliar with these terms and concepts, hindering their comprehension of the paper.

[1] Jordan T Ash, Chicheng Zhang, Akshay Krishnamurthy, John Langford, and Alekh Agarwal. Deep batch active learning by diverse, uncertain gradient lower bounds. arXiv preprint arXiv:1906.03671, 2019.

**Questions:**

a. What's the method's performance on larger datasets such as CelebA?

b. How does the proposed method compare with state-of-the-art general active learning methods in the multi-model setting?

---

> ### Author Response · Authors · 2023-11-20
>
> > What's the method's performance on larger datasets such as CelebA?
>
> Thank you for your comments. We have tested the compared methods in the CelebA dataset. The results of the performance comparisons are supplemented to Figure 3 in the revised paper.
> We can observe that our method is consistently better than the other baselines in this dataset.
>
> > How does the proposed method compare with state-of-the-art general active learning methods in the multi-model setting?
>
> Thanks for your comment. There have indeed been many deep AL methods proposed in recent years. However, the setting of AL for multi-model was first proposed by Tang and Huang [1] in NeurIPS 2022. To the best of our knowledge, DIAM [1] is the only method that focuses on this setting so far. We have also considered comparing with other deep AL algorithms; however, it is non-trivial to adapt existing methods to the multi-model setting. We follow most of the experimental settings in [1], including the compared baselines, such as Uncertainty, Coreset and so on.
>
> > The paper's writing, particularly in Section 3.1, may benefit from further refinement.
>
> We have added some explanatory text to Section 3.1. We shall make further refinements in the next version.
>
> [1] Ying-Peng Tang and Sheng-Jun Huang. Active learning for multiple target models. In Advances in Neural Information Processing Systems, 2022.

---

> > ### Comment · Reviewer_pbqA · 2023-11-23
> >
> > Thanks for your response! I will keep my score.

---

### Official Review · Reviewer_KBY8 · 2023-11-08

**Soundness:** 3 good
**Presentation:** 3 good
**Contribution:** 2 fair
**Rating:** 5
**Confidence:** 3

**Summary:**

This paper proposed a one-shot active learning algorithm which only queries the unlabelled target data once, which is in contrast to the existing AL algorithms that often rely on iterative model training. This one-shot query can reduce computation costs from repeated training, especially for deep models. Specifically, the authors extract representations of the same dataset using distinct network backbones and learn the linear prediction layer on each representation via an $\ell_p$-regression formulation.

**Strengths:**

This paper tackles the one-shot active learning, which is an interesting idea since the iterative training procedure of active learning can indeed be computationally expensive. The proposed method is theoretically rooted and technically sound. Besides, the source code showed that the proposed algorithm seems to be reproducible.

**Weaknesses:**

My primary concern is about the problem setting. As a motivation, the authors claim that the iterative training of traditional active learning can be computationally expensive, and the same instance can have different representations through distinct backbones. However, in the proposed method, the dataset was pre-processed with 50 different models to get the representation, which, in my view, is still computationally expensive. Thus, I found the motivation for one-shot active learning is somehow weak.

My second concern was about the empirical evaluations. The baselines are compared in the empirical evaluations. Only DIAM was published in the past two years. Some recent Deep AL method is missing in the empirical evaluation.

Furthermore, the benchmark datasets evaluated in the paper are somehow simple. More complex and challenging dataset, e.g. CIFAR100, which was widely compared in the AL literature, should also be tested.

**Questions:**

I have some questions and comments, and I hope the authors can try to clarify.

1. My first question is about the one-shot query. I do admit that it will reduce the computational burden; however, I'm wondering whether the performance will be comparable with the traditional iterative training procedure.
2. In the empirical setting part, I'm confused about the setting of *50 distinct architectures*. Since it's noted in the footnote that all the experiments were conducted with the same GPU and CPU, I'm wondering whether it's still necessary to employ such method?
3. As I mentioned in the weakness part, more recent baselines should be compared to demonstrate the effectiveness of the proposed method.

---

> ### Author Response · Authors · 2023-11-20
>
> > The proposed method is still computationally expensive, as it pre-processes the dataset with 50 different models to get the representation. Besides, it is better to validate whether the performance will be comparable with the traditional iterative training procedure.
>
> Thanks for your comment. To the best of our knowledge, most existing AL methods, such as Uncertainty [1], learning loss [2], batchBALD [3], BADGE [4], require feeding the unlabeled data into the target network at least once to gather necessary information for data selection. Under the setting of multiple target models studied in this work, we believe it is necessary to feed the data into each target network in order to estimate the preferences of different models. Moreover, the time cost of this process is much less than that of iteratively training multiple deep models. Therefore, we hold the point that our method is efficient for multiple models.
>
> The results of the comparisons between our method and the traditional iterative methods are presented in Figure 2 of the paper. It can be observed that our method can achieve comparable or even better performances when compared to the iterative methods. These results indicate that our method can significantly reduce the costs of training multiple deep models while saving the number of queries.
>
> > I'm confused about the setting of 50 distinct architectures. Since it's noted in the footnote that all the experiments were conducted with the same GPU and CPU, I'm wondering whether it's still necessary to employ such method?
>
> We employ 50 distinct network architectures in our empirical studies to simulate the learning scenario of adapting the machine learning system to multiple devices. Our goal is to reduce the sample complexity using one-shot querying. To compare the sample complexity with the traditional iterative methods, we run all the compared methods on the same GPU and CPU simply for fair comparisons. We do not measure the running times of the methods on machines with those architectures. In real-world applications, it is easy to train different models in parallel on multiple machines.
>
> > In the experiment section, more recent baselines should be compared to demonstrate the effectiveness of the proposed method. More complex and challenging dataset, e.g. CIFAR100, which was widely compared in the AL literature, should also be tested.
>
> Thank you for your comments. We have tested the compared methods in the CIFAR100 and EMNIST-digits datasets. The results of the performance comparisons are supplemented to Figure 2 in the revised paper. We can observe that our method is significantly better than the other baselines on these datasets.
>
> As to the recent baselines, we note that the setting of AL for multi-model was first proposed by Tang and Huang [5] in NeurIPS 2022. To the best of our knowledge, DIAM [5] is the only method that focuses on this setting so far. We have also considered comparing with other deep AL algorithms; however, it is non-trivial to adapt existing methods to the multi-model setting. We follow most of the experimental settings in [5], including the compared baselines, such as Uncertainty, Coreset and so on.
>
>
>
> [1] David D. Lewis and William A. Gale. A sequential algorithm for training text classifiers.
> In W. Bruce Croft and C. J. van Rijsbergen, editors, Proceedings of the 17th Annual International ACM-SIGIR Conference on Research and Development in Information Retrieval,
> pages 3–12. ACM/Springer, 1994.
>
> [2] Donggeun Yoo and In So Kweon. Learning loss for active learning. In Proceedings of the
> IEEE/CVF conference on computer vision and pattern recognition, pages 93–102, 2019.
>
> [3] Andreas Kirsch, Joost Van Amersfoort, and Yarin Gal. Batchbald: Efficient and diverse
> batch acquisition for deep bayesian active learning. Advances in neural information processing systems, 32, 2019.
>
> [4] Jordan T Ash, Chicheng Zhang, Akshay Krishnamurthy, John Langford, and Alekh Agarwal. Deep batch active learning by diverse, uncertain gradient lower bounds. 2020.
>
> [5] Ying-Peng Tang and Sheng-Jun Huang. Active learning for multiple target models. In
> Advances in Neural Information Processing Systems, 2022.

---

### Author Response · Authors · 2023-11-20

We sincerely thank the reviewers for the efforts and the constructive comments. We have carefully revised the paper accordingly. In the following, we address the concerns and answer the questions raised by the reviewers. Please also see the revised paper (has been uploaded in OpenReview).

---

### Meta-Review · Area_Chair_9Gjm · 2023-12-11

**Metareview:**

The paper tackles the problem of reducing the computational burden and label complexity of active learning multiple models simultaneously on the same data. The approach taken views each model as a nonlinear feature extractor followed by a linear prediction layer, and frames active learning as choosing a shared sampling matrix that minimizes the prediction error across the models simultaneously. Assuming that the errors used in fitting the linear prediction layer of each model is lp-regression, the authors devise an algorithm that samples unlabeled observations by the maximum of the Lewis weights of their nonlinear representations across all the models, and establish an approximation guarantee for the algorithm. Empirical results are provided on classification and regression data sets, showing that the novel method outperforms the previous state-of-the-art algorithm for this problem, and prior active learning approaches adapted to the multimodel setting.

The empirical evaluations are limited in that they do not compare to recent active learning algorithms, due to the difficulty of adapting them to the multiple model setting; it is also not clear how the method is employed in the experimental evaluations when the model is learned from scratch, or when the loss is a classification loss. Nonetheless, the algorithm is theoretically well-founded, and the empirical results demonstrate that it outperforms the previous SOTA algorithm for the same problem.

**Justification For Why Not Higher Score:**

The empirical evaluations are limited in that they do not compare to recent active learning algorithms, due to the difficulty of adapting them to the multiple model setting; it is also not clear how the method is employed in the experimental evaluations when the model is learned from scratch, or when the loss is a classification loss.

**Justification For Why Not Lower Score:**

The paper presents a solution to the multimodel active learning problem that is theoretically well-founded, and outperforms the current SOTA algorithm.

---

### Decision · Program_Chairs · 2024-01-16

Accept (poster)